# Topology Explains Loss Barriers: Quotient Homology of Neural Loss Landscapes

## Abstract

Although neural network parameter spaces are contractible, training exhibits global phenomena that elude purely Euclidean accounts. We show that these effects arise up to symmetry after factoring out ubiquitous reparameterizations, the low-loss regions of the quotient landscape acquire nontrivial homology. For semialgebraic losses and common symmetry groups, we prove that quotient sublevel sets $S_c/G$ have finite Betti numbers and that $\beta_k(S_c/G) > 0$ yields a topological certificate of barriers. Also, we operationalize these insights with symmetry-aware trajectories (permutation alignment, scale normalization, Stiefel-consistent updates) that remove spurious obstacles and expose genuine connectivity in the quotient space. Experiments on Stiefel-constrained autoencoders and residual networks support the theory: homology summaries of sublevel sets predict the presence or absence of interpolation barriers, and quotient-aware paths recover robust mode connectivity. Taken together, our results provide a principled and testable account of why weight matching works and when loss barriers are intrinsic rather than artifacts of parameter redundancy.

## 1 Introduction

Modern neural networks operate in extremely high-dimensional parameter spaces that are, from a purely topological standpoint, simple and contractible Euclidean spaces. Yet empirical and theoretical studies show that training dynamics and loss geometry exhibit rich global phenomena, including connected low-loss corridors between independently trained solutions, apparent loss barriers, and large families of functionally equivalent models (Li et al. (2018); Garipov et al. (2018); Frankle et al. (2020b)). These observations suggest that the ambient Euclidean view is insufficient to explain global organization of solutions.

A growing body of work points to symmetry as a primary driver of this structure. Weight-space permutation invariance and related reparameterizations create large equivalence classes of parameters that can be rearranged to uncover low-loss paths between optima (mode connectivity), as well as surprisingly flat behavior along straight-line interpolation (linear mode connectivity) once appropriate alignment has been performed (Entezari et al. (2021); Brea et al. (2019); Ainsworth et al. (2023)). Recent analyses make this link explicit, showing that both mode connectivity and linear mode connectivity often emerge only after solving a weight-matching or alignment problem that factors out neuron permutations and scalings (Frankle et al. (2020a); Entezari et al. (2022)), with extensions to specialized architectures such as graph neural networks and updated perspectives emphasizing symmetry as an organizing principle (Li et al. (2025); Zhao et al. (2025b)). Beyond individual case studies, visualization and profiling tools consistently reveal broad valleys and connected basins when models are compared up to symmetry (Li et al. (2018)).

These empirical findings motivate a topological perspective on low-loss regions that treats symmetries via quotient spaces. In this view, one studies sublevel sets of the loss composed with the canonical projection onto equivalence classes, and asks for global invariants (e.g., homology and homotopy groups) of these symmetry-quotiented sets. Topological data analysis has begun to quantify such structure directly from trained models and landscapes, relating the presence of nontrivial features to generalization and robustness across architectures, including ResNets and physics-informed networks (Horoi et al. (2021); Xie et al. (2024); Geniesse et al. (2025)). In parallel, Morse-theoretic analyses provide a bridge between critical-point structure and connectivity properties of the land-

scape (Akhtiamov & Thomson (2023)), while Betti-number summaries offer a coarse but computable description of the shape of loss surfaces and their low-loss regions (Bucarelli et al. (2024)). Our framework formalizes these ideas using classical tools from algebraic topology like singular homology, homotopy groups, and duality theorems in the symmetry-quotiented setting (Hatcher (2000)), and it connects them to categorical and homotopy-theoretic viewpoints on neural systems more broadly (Manin & Marcolli (2023)).

Concretely, we show that accounting for natural neural symmetries induces nontrivial topology in quotient low-loss regions, even though the ambient parameter space is contractible. Nonvanishing Betti numbers of a quotient sublevel set $S_c/G$ signal essential cycles and homological obstructions. Via Alexander duality, they certify that there exist solutions whose images in $S_c/G$ cannot be joined by any continuous path that stays within loss $\leq c$, so any such path must somewhere incur loss $> c$. This perspective unifies and sharpens prior empirical observations of linear/curvilinear mode connectivity (Garipov et al. (2018); Frankle et al. (2020b); Li et al. (2025); Zhao et al. (2025a)), the central role of permutation invariance (Entezari et al. (2021); Brea et al. (2019)), and the emerging use of topological summaries for landscape analysis (Xie et al. (2024); Geniesse et al. (2025); Horoi et al. (2021)), while grounding the discussion in rigorous algebraic-topological foundations and complementary homotopy-theoretic formalisms (Manin & Marcolli (2023)). Finally, we provide practical guidelines for symmetry-aware visualization and profiling (Li et al. (2018)), showing how quotient-aware analyses compress parameter spaces and clarify when barriers are genuine topological obstructions rather than artifacts of misaligned coordinates (Zhao et al. (2025b)).

## 1.1 RELATED WORK

Recent research has increasingly focused on understanding the complex topology and geometry of neural network loss landscapes. This line of work aims to explain observed phenomena such as mode connectivity, loss barriers, and the presence of flat and sharp minima in high-dimensional parameter spaces.

**Topology and Geometry of Neural Network Loss Landscapes.** Several studies have employed tools from topological data analysis (TDA) and algebraic topology to examine the global structure of loss surfaces. For example, Horoi et al. (2021) used manifold-preserving dimensionality reduction combined with computational homology to analyze loss landscapes, linking topological signatures to generalization performance. Building on this, Xie et al. (2024) utilized Betti numbers to characterize loss landscape topology across different architectures, demonstrating how these invariants elucidate learning dynamics.

**Mode Connectivity and Symmetry in Neural Networks.** Mode connectivity, the phenomenon where low-loss paths connect distinct minima, has been explained partly by underlying parameter space symmetries Zhao et al. (2025a); Ainsworth et al. (2023) highlighted the role of permutation and scaling symmetries in uncovering connectedness between modes. In Li et al. (2025) the authors investigated these effects in graph neural networks, revealing non-linear connectivity shaped by graph structure, emphasizing the importance of quotienting by symmetries to understand loss barriers.

**Permutation Symmetry and Loss Landscape Structure.** Permutation symmetry in neural network layers introduces multiple equivalent minima and saddle points, complicating optimization landscapes. Brea et al. (2019) rigorously studied permutation symmetries and demonstrated the arising of permutation saddles and flat valleys. Their work supports the idea that quotienting parameter spaces by neuron permutations is crucial to reveal the true topology impacting training trajectories.

**Mathematical Foundations: Homotopy and Homology in Neural Contexts.** Foundational work by Manin & Marcolli (2023) applied homotopy-theoretic and categorical frameworks to neural information networks, linking homotopy types to function representation and network coding. These theoretical insights underpin the mathematical formalism used to analyze topological obstructions in low-loss regions and justify the use of homology to certify mode connectivity barriers.

**Visualization and Quantification of Loss Landscapes.** Early work by Li et al. (2018) introduced visualization methods illustrating the impact of sharpness on generalization. More recent approaches, such as the topological landscape profiles developed by Geniesse et al. (2025), leverage persistence diagrams to quantify higher-dimensional topological features correlated with training stability and model robustness.

Together, these studies develop a rigorous topological and geometric perspective on neural network training landscapes, complementing the homotopy and homology analyses presented in this paper. They collectively highlight the critical role of symmetry quotients and topological invariants in explaining optimization paths, mode connectivity, and loss barriers.

## 1.2 CONTRIBUTIONS

Conceptually, our theoretical contribution has two layers. First, we prove general structural results showing that, under mild assumptions, quotient sublevel sets $S_c/G$ are well-behaved objects with finite Betti numbers, so that homology provides meaningful global descriptors of the low-loss region modulo symmetry. Second, we exhibit explicit constrained architectures where these Betti numbers are provably nonzero at small loss levels, demonstrating that intrinsic barriers can indeed occur even after factoring out symmetries. Specifically,

- For semi-algebraic losses and finite symmetry groups, sublevel sets and their quotients have finite Betti numbers; whenever $\beta_k(S_c/G) > 0$, there exists a noncontractible $k$-cycle whose closed geometric support has a small neighborhood $U$ such that two minima lie in different path components of $(S_c/G) \setminus U$.

- We show how natural constraints and reparameterization symmetries change topology: (i) rescaling quotients can create sphere factors with nontrivial $\pi_1$; (ii) unit-row-norm constraints yield products of spheres with nontrivial higher homotopy, implying intrinsic homotopic obstructions invisible in the unconstrained Euclidean view.

- Lower bounds on $\beta_{k-1}(S_c/G)$ imply lower bounds on the number of distinct path components of the complement of small neighborhoods of $(k-1)$-dimensional cycles in $S_c/G$. In particular, they certify the existence of multiple low-loss regions in the quotient that cannot be merged without crossing loss $> c$.

- Case studies demonstrate that symmetry-aligned homotopies remove spurious barriers and recover smooth low-loss paths within $S_c/G$.

Our results establish a rigorous topological foundation to explain and anticipate observed phenomena such as mode connectivity, sharp versus flat minima, and the multiplicity of functionally equivalent solutions in modern deep learning. By shifting the focus from the ambient parameter space to the topology of symmetry-quotiented low-loss regions, this work opens new avenues for understanding optimization landscapes, guiding architecture design, and improving training strategies based on topological invariants.

## 2 PRELIMINARIES

We only require a small amount of algebraic topology. At a high level, we use singular homology to detect holes in low-loss regions, carriers and geometric supports to talk about where a homology class lives in the parameter space, and Alexander duality to relate holes in a sublevel set to the connectivity of its complement. We may think of a singular 1–simplex as a continuous parametrized curve, a 1–cycle as a closed loop with no boundary, and the homology group $H_k(X; G)$ as describing $k$–dimensional holes in a space $X$ with coefficients in an abelian group $G$. We briefly recall the relevant definitions below and refer to Appendix A for background.

**Neural Network Parameter Spaces.** We consider feed-forward neural networks of fixed architecture and depth $L$, with weight and bias parameters collected into a vector $\theta = \left(W^{(1)}, b^{(1)}, \ldots, W^{(L)}, b^{(L)}\right) \in \mathbb{R}^D$, where $D = \sum_{\ell=1}^{L} \left(n_\ell n_{\ell-1} + n_\ell\right)$ is the total number of scalar

parameters. We denote $\mathbb{F}(n, m, d) = \mathbb{R}^D$ as the ambient parameter space of a network mapping $\mathbb{R}^n \to \mathbb{R}^m$ with piecewise-linear activations in $\mathbb{R}^d$.

**Geometric Support.**

**Definition 2.1.** *Let $X$ be a topological space and let $z = \sum_{i=1}^N a_i \sigma_i$ be a singular $k$-chain on $X$ with $a_i \in \mathbb{Q}$ and each $\sigma_i \colon \Delta^k \to X$ continuous. The geometric support of $z$ is*

$$|z| = \bigcup_{i \colon a_i \neq 0} \sigma_i(\Delta^k) \subset X,$$

*and the closed support of $z$ is $\overline{|z|}$, the closure in $X$.*

Intuitively, a singular $k$–simplex $\sigma \colon \Delta^k \to X$ is just a continuous parametrized $k$–dimensional patch in $X$, and a chain $z$ is a formal linear combination of such patches. The geometric support $|z|$ is the union of all patches that appear with nonzero weight; this is the subset of $X$ where the chain lives.

**Definition 2.2.** *Let $G$ be an abelian group and let $\alpha \in H_k(X; G)$. A closed subset $A \subset X$ is called a carrier of $\alpha$ if the inclusion $i \colon A \hookrightarrow X$ satisfies $\alpha \in \mathrm{im}\big(H_k(A; G) \xrightarrow{i_*} H_k(X; G)\big)$. The closed geometric support of $\alpha$ is the intersection of all closed carriers*

$$|\alpha| = \bigcap \Big\{ A \subset X \text{ closed} \colon \alpha \in \mathrm{im}\big(H_k(A; G) \to H_k(X; G)\big) \Big\}.$$

*Then $|\alpha|$ is closed and is the minimal closed subset of $X$ that supports $\alpha$ in the above sense.*

A carrier of a homology class $\alpha$ is therefore a closed set on which $\alpha$ can be represented by a cycle, and the closed geometric support $|\alpha|$ is the smallest such closed set.

**Remark 2.3.** *Given a cycle representing a class $\alpha \in H_k(X; G)$, any sufficiently small open neighborhood $U$ of the closed geometric support $|\alpha|$ contains the geometric support of some cycle representative of $\alpha$. Consequently, separation arguments that use the complement $X \setminus U$ are independent of the particular representative chosen for $\alpha$.*

**Semi-algebraic sets**

**Definition 2.4.** *A subset $S \subseteq \mathbb{R}^n$ is called semi-algebraic if it can be written as a finite union of sets of the form $\{x \in \mathbb{R}^n \colon f_1(x) = 0, \ldots, f_r(x) = 0, g_1(x) > 0, \ldots, g_s(x) > 0\}$ where $f_i, g_j \in \mathbb{R}[x_1, \ldots, x_n]$ are real polynomials. A map $F \colon S \to \mathbb{R}^m$ defined on a semi-algebraic set $S \subseteq \mathbb{R}^n$ is called semi-algebraic if its graph is a semi-algebraic subset of $\mathbb{R}^{n+m}$.*

It is important to note that compact semi-algebraic sets admit finite triangulations and have the homotopy type of finite CW complexes, ensuring finite Betti and cohomology groups.

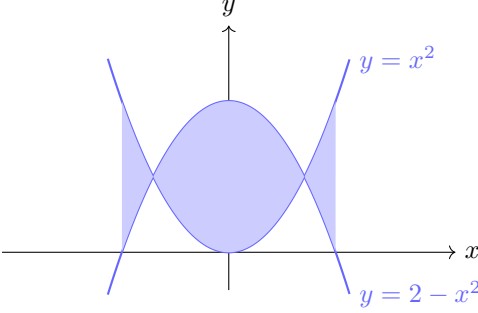

Figure 1: A semi-algebraic set given by $x^2 \leq y \leq 2 - x^2$.

More details in Appendix A.

## 3 THEORETICAL RESULTS

Let $\mathcal{F}$ be the actual parameter space of a neural network architecture of fixed width and depth. In the usual setting, $\mathcal{F}$ is homeomorphic to $\mathbb{R}^d$, and thus is contractible. Consequently, its fundamental group $\pi_1(\mathcal{F})$ is trivial, so there are no nontrivial loops arising purely from the parameterization.

This section unfolds in two steps. We first establish general properties of quotient sublevel sets $S_c/G$ under mild regularity assumptions, culminating in Theorem 3.4, which guarantees finiteness of Betti numbers and a transfer relation between homology of $S_c$ and $S_c/G$ for finite groups. We then specialize to concrete constrained architectures with unit-norm rows and Stiefel factors, for which we can compute or bound the homotopy type of $F_{\mathrm{norm}}$ and thereby show that $\beta_k(S_c/G) > 0$ for some $k \geq 1$ at sufficiently small loss levels.

All proofs can be found in Appendix B.

**Proposition 3.1.** *Let $L \geq 1$ be a fixed depth and let $(n_0, n_1, \ldots, n_L)$ be the widths of each layer in a fully-connected feed-forward neural network with continuous activation functions. Define the parameter space*

$$\mathcal{F} = \prod_{\ell=1}^{L} \big( \mathbb{R}^{n_\ell \times n_{\ell-1}} \times \mathbb{R}^{n_\ell} \big),$$

*so that each parameter $\theta$ consists of weight matrices $W^{(\ell)} \in \mathbb{R}^{n_\ell \times n_{\ell-1}}$ and bias vectors $b^{(\ell)} \in \mathbb{R}^{n_\ell}$. Then $\mathcal{F} \cong \mathbb{R}^d$, for some $d$ and hence $\mathcal{F}$ is contractible.*

When such loops arise, they may signal subtleties in the loss landscape or potential obstructions in navigating between different global minima.

Studying contractibility in the context of neural network parameter spaces is valuable because it ensures path-connectedness. It implies that the ambient parameter space has trivial homotopy groups, so any nontrivial topology relevant for barriers must arise from the geometry of the loss function (through the topology of its sublevel sets) rather than from the domain itself. And once one quotients out natural reparameterization symmetries, one can isolate the nontrivial topology caused solely by redundancies rather than by the space itself.

Nonetheless, the existence of nontrivial loops in $\widetilde{\mathcal{F}}$ does not automatically imply failure of gradient-based methods to find good solutions in practice. Local geometry, saddle points, and other analytic factors also strongly influence the behavior of gradient descent.

Neural networks often admit nontrivial symmetries. For example, in a one-hidden-layer network with ReLU one can simultaneously scale the first-layer weights and rescale the second layer:

$$(W^{(1)}, b^{(1)}, W^{(2)}, b^{(2)}) \sim (\lambda W^{(1)}, \lambda b^{(1)}, \lambda^{-1} W^{(2)}, b^{(2)}), \quad \lambda > 0.$$

Define an equivalence relation $\sim$ on $\mathcal{F}$ by these rescalings, and consider the quotient space $\widetilde{\mathcal{F}} = \mathcal{F}/\sim$. Equipped with the quotient topology, $\widetilde{\mathcal{F}}$ need no longer be contractible. In fact we can show when $\widetilde{\mathcal{F}} \cong \mathbb{S}^{n_1(n_0+1)-1} \times \mathbb{R}^{n_2(n_1+1)}$ nontrivial loops appear if and only if $n_1(n_0 + 1) = 2$ (see Proposition B.3 for an analogous statement in a simplified linear setting).

Thus although the raw parameter space $\mathcal{F}$ is contractible, quotienting by the natural scaling symmetry can introduce nontrivial topology, with implications for the landscape of equivalent models under reparametrization.

Since the ambient parameter space is contractible, any local minima or saddles encountered during training are created by the loss landscape itself; there are no purely topological obstructions coming from the parameter domain before the loss is applied.

**Proposition 3.2.** *Let $p \geq 3$ and $m \geq 1$. Consider a single-layer neural network $f_\theta \colon \mathbb{R}^p \to \mathbb{R}^m$, $x \mapsto Wx + b$, where $W \in \mathbb{R}^{m \times p}$ and $b \in \mathbb{R}^m$. Suppose that for each $1 \leq j \leq m$, $\|W_{j,*}\| = 1$, where $W_{j,*} \in \mathbb{R}^p$ is the $j$-th row of $W$ and $\|\cdot\|$ is the Euclidean norm. Denote by $\mathcal{F}_{norm}(p, m)$ the set of all such $(W, b)$ satisfying these constraints. Then $\mathcal{F}_{norm}(p, m)$ has nontrivial higher homotopy groups.*

Row norm constraints arise as simplified models of practical architectural choices such as weight normalization and spectral or orthogonality regularization, which are used in modern architectures including orthogonal RNNs and normalized convolutions. Even after quotienting out reparameterization symmetries, these constraints turn the parameter space $F_{\text{norm}}(p, m)$ into a product of spheres and Euclidean factors rather than a contractible Euclidean space. The nontrivial homotopy of $F_{\text{norm}}$ therefore captures structural obstructions to parameter homotopies that are invisible in unconstrained settings and has direct implications for mode connectivity and loss barriers.

Let's now extend the Proposition 3.2

**Theorem 3.3.** *Let $L \geq 1$ and widths $(n_0, n_1, \ldots, n_L)$ with $n_0 = p$ and $n_L = m$. Consider the parameter space*

$$\mathcal{F}_{\text{norm}}(n_0, \ldots, n_L) = \prod_{\ell=1}^{L} \left\{ (W^{(\ell)}, b^{(\ell)}) \colon W^{(\ell)} \in \mathbb{R}^{n_\ell \times n_{\ell-1}}, b^{(\ell)} \in \mathbb{R}^{n_\ell}, \|W_{j,*}^{(\ell)}\| = 1, 1 \leq j \leq n_\ell \right\}.$$

*Then there is a canonical homeomorphism $\mathcal{F}_{\text{norm}}(n_0, \ldots, n_L) \cong \left( \prod_{\ell=1}^{L} \left( \mathbb{S}^{n_{\ell-1}-1} \right)^{n_\ell} \right) \times \mathbb{R}^{\sum_{\ell=1}^{L} n_\ell}$. In particular, for any layer index $\ell$ with $n_{\ell-1} \geq 3$ and $n_\ell \geq 1$, $\pi_{n_{\ell-1}-1}\left( \mathcal{F}_{\text{norm}}(n_0, \ldots, n_L) \right) \cong \mathbb{Z}^{n_\ell}$.*

Now let $X \subset \mathbb{R}^D$ be a compact semialgebraic set, let a finite group $G$ act on $X$ by semialgebraic homeomorphisms, and let $L \colon X \to \mathbb{R}$ be $G$-invariant and semialgebraic. For $c \in \mathbb{R}$ set $S_c = L^{-1}((-\infty, c])$ and write $\pi \colon S_c \to S_c/G$ for the quotient map. Assume the following: **(A1)** There exists $R > 0$ such that $S_c \subseteq \{\|\theta\| \leq R\}$. **(A2)** Also $G$ is generated by neuron permutations in each hidden layer. Then $G$ acts continuously on $\mathcal{F}$, preserving $L$ and $S_c$. **(A3)** Both $L$ and the action of $G$ are semi-algebraic maps, so $S_c$ and the quotient $S_c/G$ are semi-algebraic sets.

The previous assumptions cover the symmetry patterns we use in our case studies. In particular, in Examples 4.2 and 4.3 the group $G$ is generated by neuron/channel permutations in each hidden layer, so $G$ is finite and acts continuously on the parameter space $F$, preserving the loss and hence each sublevel set $S_c$. The standard architectures we consider also yield semi-algebraic losses. Thus Theorem 3.4 applies directly to the permutation symmetries present in our experiments.

**Theorem 3.4.** *Under assumptions (A1)–(A3) above: (1) Each sublevel set $S_c$ is a compact, semi-algebraic subset of $\mathbb{R}^D$, and hence has finite Betti numbers $\beta_k(S_c)$ for each $k \geq 0$. (2) The quotient space $S_c/G$ is also compact and semi-algebraic, so its Betti numbers $\beta_k(S_c/G)$ are finite. And, (3) if $\beta_k(S_c/G) > 0$ for some $k \geq 1$, then there exists a nontrivial class $0 \neq [z] \in H_k(S_c/G; \mathbb{Q})$. Let $\text{tr} \colon H_k(S_c/G; \mathbb{Q}) \to H_k(S_c; \mathbb{Q})$ be the transfer map (see Proposition A.3). Then $\widetilde{z} \colon = \text{tr}([z]) \in H_k(S_c; \mathbb{Q})^G$ satisfies $\pi_*(\widetilde{z}) = [z] \neq 0$. In particular $\widetilde{z} \neq 0$, so any singular $k$–cycle representing $\widetilde{z}$ is noncontractible in $S_c$.*

**Remark 3.5.** *Theorem 3.4 is stated for finite groups, which is the natural setting for neuron/channel permutations. Parts (1)–(2) can be extended to compact Lie group actions. If a compact Lie group $G$ acts smoothly and properly on a compact triangulable set $S_c$, then the quotient $S_c/G$ is again a compact triangulable space with finite-dimensional singular homology. The transfer statement in part (3), which identifies a nontrivial class in $H_k(S_c/G; \mathbb{Q})$ with a $G$-invariant class in $H_k(S_c; \mathbb{Q})$, is genuinely finite-group–specific.*

Topology captures the global shape of low-loss regions that local quantities may miss. Theorem B.19 formalizes the intuition that the optimizer travels through a landscape whose nearby low-loss geometry is stable except at isolated, predictable bifurcation times and each such bifurcation has a clean topological signature (the attachment of a cell of dimension equal to the Morse index) that can be interpreted as the opening or closing of new paths, loops, or higher-dimensional families of models during training. For more details and examples see Appendix B.

## 4 NUMERICAL ANALYSIS

For this section, we need the following: we write $\text{Stiefel}(k, n)$ for the Stiefel manifold of $k \times n$ matrices with orthonormal rows. It is a smooth, compact manifold that can be endowed with a Riemannian structure and standard retractions, for instance via a thin QR factorization. Given $W \in$

$\mathbb{R}^{k \times n}$ close to $\mathrm{Stiefel}(k, n)$, we let $\mathrm{Retr}(W)$ denote the $Q$ factor in the QR decomposition of $W^\top$, transposed back to a $k \times n$ matrix. In our experiments, a Stiefel retraction perturbation of a point $W \in \mathrm{Stiefel}(k, n)$ means adding a small Gaussian tangent perturbation at $W$ and mapping back to the manifold via this retraction.

Our numerical examples deliberately use constrained models that are stylized but closely related to techniques used in practice. The Stiefel-constrained autoencoder in Example 4.2 models architectures with orthogonality priors, while the mini-ResNet in Example 4.3 reflects the finite neuron/channel permutation symmetry present in modern convolutional networks together with batch-normalization. In both cases the goal is to illustrate how architectural constraints and symmetry quotienting jointly shape the topology of low-loss regions, rather than to propose the exact architectures as practical benchmarks.

**Remark 4.1.** *Our structural results are stated for semi-algebraic losses and group actions. In practice, components such as batch normalization and adaptive optimizers introduce additional state variables and dataset dependence. In our setting, these are handled as follows: BatchNorm running means and variances are either frozen or recomputed from a fixed calibration set via finite-sample averages and affine rescaling, and optimizer states can be incorporated as extra coordinates in an extended parameter space.*

**Example 4.2.** *Let $\{x_i\}_{i=1}^N \subset \mathbb{S}^2 \subset \mathbb{R}^3$ be points sampled densely on the unit sphere, and consider the autoencoder $E_\theta(x) = W_2 \tanh(W_1 x)$, $W_1, W_2 \in \mathrm{Stiefel}(3, 2)$, with reconstruction loss $L(\theta) = \frac{1}{N} \sum_i \|E_\theta(x_i) - x_i\|^2$. In addition to the continuous $\mathrm{O}(2)$ gauge acting on the hidden 2-dimensional subspace, the architecture also inherits the usual finite neuron-permutation symmetries as in Assumption (A2). For the latter, the assumptions of Theorem 3.4 are satisfied, so for sufficiently small $c > 0$ the quotient sublevel set $S_c/G$ has $\beta_2(S_c/G) \geq 1$, detecting a noncontractible 2-sphere's worth of nearly isometric encoder–decoder parameters. The continuous $\mathrm{O}(2)$ gauge can be viewed as enriching these permutation orbits to an $\mathrm{O}(2)$-family, but the topological certificate already arises at the level of the finite quotient.*

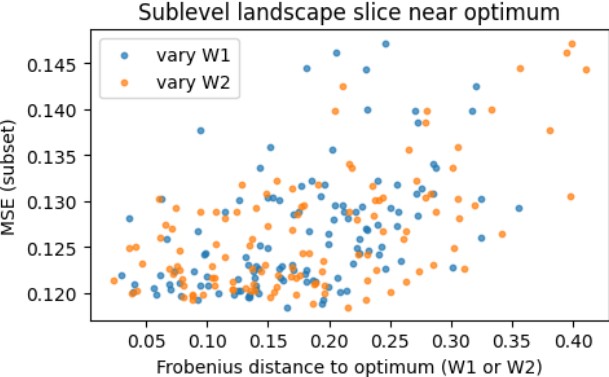

Figure 2: Local loss under Stiefel retraction perturbations of $W_1$ or $W_2$. Points are random retractions; the upward trend indicates bowl-like curvature and the scatter reflects the $O(2)$ gauge symmetry and retraction nonlinearity, suggesting a well-conditioned landscape modulo symmetry.

*Left picture in Figure 3 displays the test points covering the sphere. The right panel shows $E_\theta(x)$ normalized to lie on $\mathbb{S}^2$. Because $E_\theta(x) \in \Pi = \mathrm{span}(W_2)$ for all $x$, the normalization maps onto $\Pi \cap \mathbb{S}^2$, which is a circle. The ring in the left is precisely this intersection; it certifies that under the current architecture the decoder cannot recover the full sphere, only a 1D subset $\mathbb{S}^1 \subset \mathbb{S}^2$ after normalization. This is a geometric fingerprint of the rank–2 linear decoder without bias. This limitation is architectural rather than transient. Because the decoder always factors through a 2-dimensional bottleneck, no amount of training can make $E_\theta$ recover the full sphere without distortion. Training dynamics therefore converge to encoder–decoder pairs that provide good reconstructions on an effectively one-dimensional subset of $S^2$ (a great circle), and the resulting nearly isometric solutions organize into a nontrivial sphere in parameter space, as predicted by the homotopy analysis.*

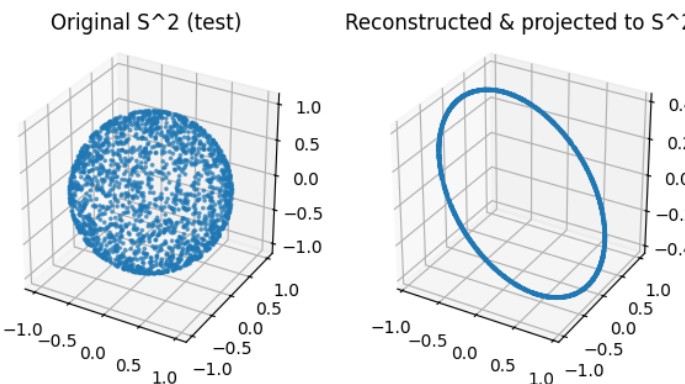

Figure 3: Stiefel-constrained autoencoder on $\mathbb{S}^2$. Left: test points on the sphere. Right: normalized decoder outputs projected to $\mathbb{S}^2$.

*The experiment intentionally constrains capacity so that the decoder output lives in a plane. The landscape slice suggests benign curvature once the $O(2)$ symmetry is factored out, motivating the study of quotient sublevel sets $S_c/G$ in Theorem 3.4.*

Theorem 3.4 reveals that the geometry of neural network training is governed not merely by the ambient Euclidean parameter space but by the topology of its low-loss regions and their symmetry-quotients. Concretely, nonzero Betti numbers of the quotient sublevel set $S_c/G$ certify the existence of noncontractible cycles in the low-loss landscape. Any continuous path joining minima or saddles whose images in the quotient lie in different path components of the complement of a small neighborhood of such a cycle must exit the sublevel set and incur higher loss. This explains why, in practice, one often observes unavoidable barriers between solutions unless one leverages nontrivial homotopies or neuron-relabeling tricks. When we speak of loss barrier, we always mean barriers at a fixed level $c$. Nontrivial homology of $S_c/G$ implies, via Alexander duality, that there exist minima or saddles whose images in the quotient lie in different path components of the complement of a small neighborhood of a cycle, so any continuous path in the quotient connecting them must leave $S_c/G$ and hence incur loss $> c$.

Lower bounds on $\beta_{k-1}(S_c/G)$ translate into lower bounds on the number of path-connected components in the complement of a representative $(k-1)$-cycle. Thus one can rigorously predict the minimal number of distinct attraction basins for gradient-based algorithms.

**Example 4.3.** *Take a mini-ResNet block with two convolutional layers and batch-norm, training on a tiny CIFAR-10 subset. Without accounting for the neuron-permutation symmetry of the channels, linear interpolation between two trained networks $\theta^A$ and $\theta^B$ often exhibits a high barrier in loss. However, if we first compute the optimal permutation $P$ of intermediate channels and re-parameterizes $\theta^B \mapsto P \cdot \theta^B$, then the quotient path lies entirely within a single path-component of $S_c/G$ and shows a monotonic low-loss interpolation. This symmetry-quotienting homotopy continuation avoids the topological barrier present in the unquotiented space and enables smooth mode connectivity. Let $\theta^A, \theta^B$ be the parameters of two independently trained mini-ResNets. We consider the linear interpolation $\theta_\alpha = (1-\alpha)\theta^A + \alpha\theta^B, \alpha \in [0,1]$, and its symmetry-aligned variant obtained by first matching hidden channels of $\theta^B$ to those of $\theta^A$ via a permutation $P$ then interpolating to $\theta_\alpha^{\mathrm{quot}} = (1-\alpha)\theta^A + \alpha(P\theta^B)$ (see Figure 4 (a)). Note that $\theta_\alpha^{\mathrm{quot}}$ traces a path inside one path-component of the quotient sublevel set $S_c/G$ (where $G$ is the channel-permutation group), thereby avoiding spurious barriers. In this setting $G$ is again the finite group of channel permutations across layers, so Assumptions (A1)–(A3) hold and Theorem 3.4 applies directly to the resulting sublevel sets $S_c$ and quotient $S_c/G$.*

*The blue and the orange curve both rise away from the endpoints, with the orange curve staying higher for most $\alpha$ and increasing near $\alpha = 1$. This behavior indicates that the aligned path, as implemented, is not function-preserving at the endpoints and that the interpolates are evaluated with inconsistent batch-normalization (BN) statistics.*

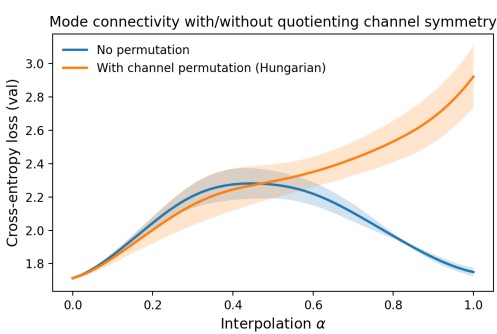 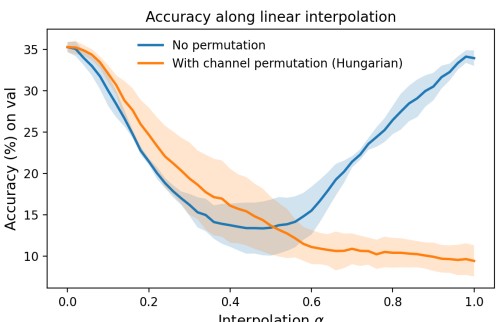

(a) Mini-ResNet on CIFAR-10: loss along interpolation between two independently trained solutions. Blue: naive linear interpolation in parameter space with fixed BN statistics. Orange: permutation-aligned interpolation using activation-based channel matching, still evaluated with fixed BN statistics.

(b) Quotient-aware interpolation (permutation alignment) combined with BN recalibration yields consistently smooth performance curves, confirming that most of the apparent barriers in the naive settings arise from symmetry mismatch and stale BN statistics rather than intrinsic topological obstructions.

Figure 4

*Now, in Figure 4 (b) we see that the accuracy drops as $\alpha$ moves into the interior of the segment, with the aligned path again underperforming. This mirrors the loss plot and is explained by the same BN issue. The two figures demonstrate that naive interpolation in networks with BN exhibits artificial barriers. After quotienting by the channel symmetry and correcting BN statistics, the interpolation typically becomes smooth and low–loss, revealing genuine mode connectivity inside $S_c/G$.*

*Algorithm 1 performs BN recalibration along the interpolation path. At each value of $\alpha$ we recompute BN running means and variances on a small calibration set and then evaluate the interpolated network with these fresh statistics. This isolates the contribution of stale BN statistics from that of the parameter interpolation itself.*

For more examples see Appendix B.

## 5 CONCLUSIONS AND FUTURE WORK

This work develops a rigorous topological framework for understanding neural network loss landscapes up to symmetry. While the ambient parameter space is contractible, natural reparameterization symmetries induce nontrivial topology in the quotient of low-loss regions. This nontrivial topology appears through nonvanishing Betti numbers and homotopy groups of quotient sublevel sets, which act as certificate-style obstructions: they show that, at a fixed level $c$, certain pairs of solutions cannot be joined by uniformly low-loss paths in the quotient.

Our analysis clarifies how such topological features relate to observed phenomena such as mode connectivity, loss barriers, and the multiplicity of functionally equivalent solutions. In particular, it explains why naive interpolation can exhibit spurious barriers and why symmetry-aware procedures often restore smooth low-loss connections. Case studies on Stiefel-constrained autoencoders and mini-ResNets concretely illustrate how quotienting by symmetries and handling BatchNorm state remove artifacts and reveal genuine connectivity.

Overall, our results are primarily structural: they shift attention from the ambient Euclidean parameter space to the topology of symmetry-quotiented low-loss sets and provide a mathematical foundation for using topological invariants as indicators of intrinsic barriers and multiple basins. We see quotient homology as complementary to local, Hessian-based TDA diagnostics and as a starting point for topology-aware architectures, regularizers, and optimization schemes. Systematically relating these invariants to concrete performance metrics and training speed remains an important direction for future work.

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

## A  ALGEBRAIC TOPOLOGY BASICS

**Homotopy, Homology and Cohomology.**   Let $(X, x_0)$ be a pointed topological space. For each integer $k \geq 0$, the $k$-th homotopy group $\pi_k(X, x_0)$ is defined as the set of based homotopy classes of continuous maps $f \colon (\mathbb{S}^k, *) \to (X, x_0)$, where $\mathbb{S}^k$ is the $k$-sphere with basepoint $*$, and two maps $f, g$ are based homotopic if there exists a continuous $H \colon \mathbb{S}^k \times [0,1] \to X$ with $H(\cdot, 0) = f$, $H(\cdot, 1) = g$, and $H(*, t) = x_0$ for all $t$. The group operation on $\pi_k(X)$ is given by concatenation of spheres (for $k \geq 1$, this makes $\pi_k(X)$ a group; for $k \geq 2$, it is abelian).

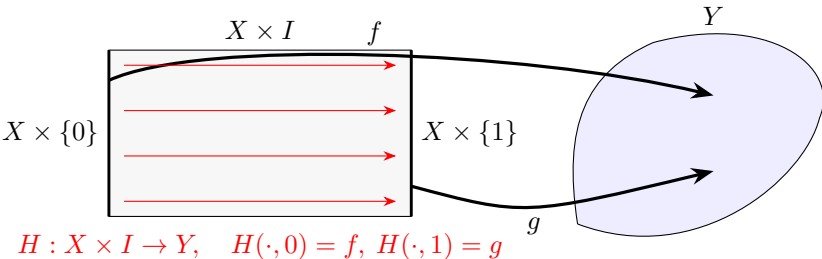

Figure 5: Homotopy between maps $f \simeq g$ as a deformation over the cylinder $X \times I$.

**Definition A.1.** *Let $X$ be a topological space and $A \subset X$ a subspace. A deformation retract of $X$ onto $A$ is a continuous map $H \colon X \times [0,1] \to X$ such that for all $x \in X$ and $a \in A$ $H(x, 0) = x$, $H(x, 1) \in A$ and $H(a, t) = a$ for all $t \in [0, 1]$. Deformation-retracting onto a subspace preserves all homotopy groups and cohomology.*

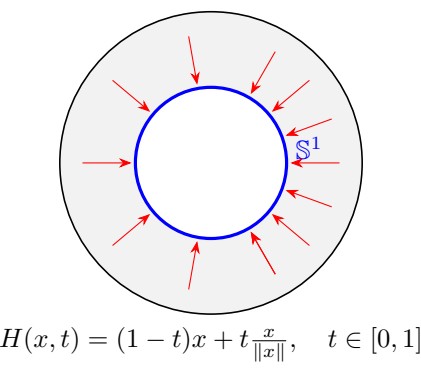

$$H(x, t) = (1-t)x + t\frac{x}{\|x\|}, \quad t \in [0, 1]$$

Figure 6: Deformation retract of an annulus onto its central circle.

**Definition A.2.** *For an abelian coefficient group $G$, the singular cochain complex of $X$ is $C^k(X; G) = \mathrm{Hom}\big(C_k(X), G\big)$, where $C_k(X)$ is the free abelian group on singular $k$-simplices $\sigma \colon \Delta^k \to X$. The coboundary operator $\delta \colon C^k(X; G) \to C^{k+1}(X; G)$ is defined by $(\delta\varphi)(\sigma) = \varphi\big(\partial\sigma\big)$, where $\partial\sigma$ is the alternating sum of the faces of $\sigma$. The $k$-th cohomology group is $H^k(X; G) = \ker(\delta \colon C^k \to C^{k+1})/\mathrm{im}(\delta \colon C^{k-1} \to C^k)$.*

When $G = \mathbb{Q}$, each $H^k(X; \mathbb{Q})$ is a finite-dimensional $\mathbb{Q}$-vector space for any compact semialgebraic set $X$ (by triangulation and the fact that $X$ has the homotopy type of a finite CW–complex). The $k$-th Betti number is $\beta_k(X) = \dim_{\mathbb{Q}} H^k(X; \mathbb{Q})$.

Throughout, let $X \subset \mathbb{R}^D$ be a compact semialgebraic set, $G$ a finite group acting on $X$ by semialgebraic homeomorphisms, and $\pi \colon X \to X/G$ the quotient map. Coefficients are over a field of characteristic 0 (we take $\mathbb{Q}$).

**Equivariant triangulation.**   By semialgebraic triangulation ( Bochnak et al. (1998); Illman (1978; 1983)) and its equivariant refinement for finite group actions on definable sets, there exists a finite

$G$–equivariant triangulation of $X$ by a simplicial complex $K$ with a simplicial $G$–action such that the quotient polyhedron $|K|/G$ triangulates $X/G$ via a simplicial map realizing $q$. Consequently,

$$C_*(X/G; \mathbb{Q}) \cong C_*(X; \mathbb{Q})_G, \tag{1}$$

where $(-)_G$ denotes coinvariants: $M_G \colon = M/\langle g \cdot m - m \rangle$.

**Proposition A.3.** *Let $G$ be finite. There is a natural chain map (the* transfer*)*

$$\mathrm{tr} \colon C_*(X/G; \mathbb{Q}) \longrightarrow C_*(X; \mathbb{Q})$$

*such that on homology*

$$q_* \circ \mathrm{tr} = \mathrm{id}_{H_*(X/G;\mathbb{Q})} \quad and \quad \mathrm{tr} \circ q_* = \mathrm{Av}_*,$$

*where* $\mathrm{Av} \colon C_*(X; \mathbb{Q}) \to C_*(X; \mathbb{Q})$ *is the Reynolds operator* $\mathrm{Av}(c) = \dfrac{1}{|G|} \displaystyle\sum_{g \in G} g_\# c$. *In particular, the induced map*

$$H_*(X/G; \mathbb{Q}) \xrightarrow{\mathrm{tr}} H_*(X; \mathbb{Q})^G$$

*is an isomorphism with inverse* $q_* \mid_{H_*(X;\mathbb{Q})^G}$, *so*

$$H_*(X/G; \mathbb{Q}) \cong H_*(X; \mathbb{Q})^G. \tag{2}$$

*Sketch of proof.* Using the $G$–equivariant triangulation, identify $C_*(X/G; \mathbb{Q})$ with $C_*(X; \mathbb{Q})_G$ via equation 1. Over $\mathbb{Q}$, invariants and coinvariants are naturally isomorphic by averaging: the canonical projection $\pi \colon C_*(X; \mathbb{Q}) \to C_*(X; \mathbb{Q})_G$ admits a $\mathbb{Q}$–linear splitting

$$s \colon C_*(X; \mathbb{Q})_G \to C_*(X; \mathbb{Q}), \quad s([c]) = \mathrm{Av}(c) = \frac{1}{|G|} \sum_{g \in G} g_\# c,$$

well defined because $[g_\# c] = [c]$ in coinvariants. Define $\mathrm{tr} \colon = s \circ$ (the isomorphism $C_*(X/G; \mathbb{Q}) \cong C_*(X; \mathbb{Q})_G$). One checks $q_\# \circ s = \mathrm{id}$ on coinvariants and $s \circ q_\# = \mathrm{Av}$ on chains, hence the stated identities on homology. Taking $G$–invariants yields equation 2. $\square$

**Remark A.4.** *No freeness is required. The argument avoids covering–space transfer and uses only that $|G|$ is invertible in $\mathbb{Q}$, so the Reynolds operator projects onto $G$–invariants. Stabilizers merely identify simplices in the quotient complex.*

**Application to quotient sublevel sets.** Let $L \colon X \to \mathbb{R}$ be semialgebraic and $G$–invariant. For $c \in \mathbb{R}$, write $S_c = L^{-1}((-\infty, c])$ and $S_c/G$ for its quotient. Both are compact semialgebraic sets with an induced finite $G$–action on $S_c$, and $q \colon S_c \to S_c/G$ is semialgebraic.

**Corollary A.5.** *For every $k \geq 0$ there is a natural isomorphism*

$$H_k(S_c/G; \mathbb{Q}) \cong H_k(S_c; \mathbb{Q})^G,$$

*with* $\mathrm{tr}$ *and* $q_*$ *inverse to each other on homology as in Proposition A.3.*

**Lemma A.6** (Barrier certificate via homology)**.** *Suppose there exists $\alpha \in H_k(S_c/G; \mathbb{Q})$ with $\alpha \neq 0$. Then any continuous map $H \colon D^{k+1} \to X/G$ whose boundary $H|_{\partial D^{k+1}}$ represents $\alpha$ must intersect the superlevel set $\{L > c\}/G$, i.e. $H(D^{k+1}) \not\subset S_c/G$. Equivalently, any such family cannot be homotoped within $S_c/G$ to a constant without crossing loss $> c$.*

*Proof.* If $H(D^{k+1}) \subset S_c/G$ then $\alpha$ would bound in $S_c/G$, contradicting $\alpha \neq 0$. Formally, $[H|_{\partial D^{k+1}}] = 0$ in $H_k(S_c/G; \mathbb{Q})$ by exactness of the pair $(D^{k+1}, \partial D^{k+1})$, so nontrivial classes cannot be filled without exiting $S_c/G$. $\square$

**Corollary A.7.** *Let $\alpha \in H_k(S_c/G; \mathbb{Q})$ be nonzero and set $\widetilde{\alpha} \colon = \mathrm{tr}(\alpha) \in H_k(S_c; \mathbb{Q})^G$. Then $\widetilde{\alpha} \neq 0$. Consequently, any $(k+1)$–parameter family $\widetilde{H} \colon D^{k+1} \to X$ with $q \circ \widetilde{H} = H$ as in Lemma A.6 must intersect $\{L > c\}$ in $X$.*

**Remark A.8.** *If one prefers to avoid Proposition A.3, one may restrict to the semialgebraic free locus $X^{\mathrm{free}} = \{x \in X \mid G_x = \{e\}\}$, which is $G$–invariant and open dense. Then $q \colon X^{\mathrm{free}} \to X^{\mathrm{free}}/G$ is a covering of degree $|G|$ and admits the classical covering–space transfer. For many applications, homology classes represented by generic cycles lie in $X^{\mathrm{free}}$, yielding the same conclusions.*

# B FULL PROOFS AND EXAMPLES

**Proposition B.1.** *Let $L \geq 1$ be a fixed depth and let $(n_0, n_1, \ldots, n_L)$ be the widths of each layer in a fully-connected feed-forward neural network with continuous activation functions. Define the parameter space*

$$\mathcal{F} = \prod_{\ell=1}^{L} \left( \mathbb{R}^{n_\ell \times n_{\ell-1}} \times \mathbb{R}^{n_\ell} \right),$$

*so that each parameter*

$$\theta = \left( W^{(1)}, b^{(1)}, W^{(2)}, b^{(2)}, \ldots, W^{(L)}, b^{(L)} \right)$$

*consists of weight matrices $W^{(\ell)} \in \mathbb{R}^{n_\ell \times n_{\ell-1}}$ and bias vectors $b^{(\ell)} \in \mathbb{R}^{n_\ell}$. Then*

$$\mathcal{F} \cong \mathbb{R}^d, \quad d = \sum_{\ell=1}^{L} \left( n_\ell n_{\ell-1} + n_\ell \right),$$

*and hence $\mathcal{F}$ is contractible. In particular,*

$$\pi_1(\mathcal{F}) = 0.$$

*Proof.* By construction, the network has depth $L$ and widths $(n_0, \ldots, n_L)$. We write

$$\mathcal{F} = \left( \mathbb{R}^{n_1 \times n_0} \times \mathbb{R}^{n_1} \right) \times \left( \mathbb{R}^{n_2 \times n_1} \times \mathbb{R}^{n_2} \right) \times \cdots \times \left( \mathbb{R}^{n_L \times n_{L-1}} \times \mathbb{R}^{n_L} \right).$$

Each factor $\mathbb{R}^{n_\ell \times n_{\ell-1}} \times \mathbb{R}^{n_\ell}$ is homeomorphic to $\mathbb{R}^{n_\ell n_{\ell-1} + n_\ell}$. Therefore

$$\mathcal{F} \cong \mathbb{R}^{\sum_{\ell=1}^{L} (n_\ell n_{\ell-1} + n_\ell)} = \mathbb{R}^d.$$

We embed $\mathbb{R}^d$ with its standard topology and $\mathcal{F}$ with the product topology. Write

$$\Phi(x) = \left( W^{(1)}(x), b^{(1)}(x), \ldots, W^{(L)}(x), b^{(L)}(x) \right)$$

where each coordinate map

$$W^{(\ell)} \colon \mathbb{R}^d \to \mathbb{R}^{n_\ell \times n_{\ell-1}}, \quad b^{(\ell)} \colon \mathbb{R}^d \to \mathbb{R}^{n_\ell}$$

is given by projection onto the appropriate block of $x$. Projections from $\mathbb{R}^d$ onto each coordinate factor are linear maps of the form

$$x \mapsto (x_{i_1}, x_{i_2}, \ldots, x_{i_k}),$$

hence continuous. Finite products of continuous maps are continuous in the product topology. Therefore

$$\Phi \colon \mathbb{R}^d \to \prod_{\ell=1}^{L} \left( \mathbb{R}^{n_\ell \times n_{\ell 1}} \times \mathbb{R}^{n_\ell} \right)$$

is continuous.

Conversely, the inverse map

$$\Phi^{-1} \colon \mathcal{F} \to \mathbb{R}^d$$

extracts each entry of the matrices $W^{(\ell)}$ and vectors $b^{(\ell)}$ and places them into a single vector in $\mathbb{R}^d$. Each extraction is again a projection in the product topology, hence continuous. Since $\Phi^{-1}$ is a finite concatenation of such projections, it is continuous. Thus $\Phi$ is a homeomorphism.

It is a standard fact that $\mathbb{R}^d$ is contractible. Pick any base point $x_0 \in \mathbb{R}^d$ and define the homotopy

$$H \colon \mathbb{R}^d \times [0,1] \to \mathbb{R}^d, \quad H(x,t) = (1-t)x + tx_0.$$

Then $H(x,0) = x$ is the identity and $H(x,1) = x_0$ is constant.

Since $\mathcal{F} \cong \mathbb{R}^d$, it inherits the same contraction. Consequently all homotopy groups of $\mathcal{F}$ are trivial. In particular,

$$\pi_1(\mathcal{F}) = 0.$$

This completes the proof. $\qquad \square$

**Example B.2.** *Consider a one-hidden-layer network ($L = 2$) with input dimension $n_0 = 1$, hidden width $n_1 = 2$ and output dimension $n_2 = 1$. Then*

$$\mathcal{F} = \left(\mathbb{R}^{n_1 \times n_0} \times \mathbb{R}^{n_1}\right) \times \left(\mathbb{R}^{n_2 \times n_1} \times \mathbb{R}^{n_2}\right) = \left(\mathbb{R}^{2 \times 1} \times \mathbb{R}^2\right) \times \left(\mathbb{R}^{1 \times 2} \times \mathbb{R}^1\right).$$

*Thus*

$$\mathcal{F} \cong \mathbb{R}^{2 \cdot 1 + 2} \times \mathbb{R}^{1 \cdot 2 + 1} = \mathbb{R}^4 \times \mathbb{R}^3 \cong \mathbb{R}^7.$$

*Explicitly, we can write*

$$\theta = \left(W^{(1)} = \begin{pmatrix} w_{11} \\ w_{21} \end{pmatrix}, b^{(1)} = (b_1, b_2), W^{(2)} = (w_{31}, w_{32}), b^{(2)} = b_3\right).$$

*The homeomorphism $\Phi : \mathbb{R}^7 \to \mathcal{F}$ simply groups the seven real coordinates $(x_1, \ldots, x_7)$ into those entries, and its inverse flattens them back. By the contraction homotopy*

$$H(x, t) = (1 - t)x + t \cdot 0$$

*in $\mathbb{R}^7$, all loops in $\mathcal{F}$ shrink to the origin. Hence $\mathcal{F}$ is contractible and $\pi_1(\mathcal{F}) = 0$ in this concrete case.*

**Proposition B.3.** *Let $n_0, n_1, n_2 \geq 0$ and consider the one–hidden–layer parameter space*

$$F = \mathbb{R}^{n_1(n_0+1)} \times \mathbb{R}^{n_2(n_1+1)} = \underbrace{\left(W^{(1)}, b^{(1)}\right)}_{v} \times \underbrace{\left(W^{(2)}, b^{(2)}\right)}_{u}.$$

*Define an equivalence relation that collapses positive rays in the first block:*

$$(v, u) \sim (\lambda v, u) \quad \textit{for all } \lambda > 0.$$

*Let $\widetilde{F} := F/\sim$ with the quotient topology. Then*

$$\widetilde{F} \cong \left((\mathbb{R}^{n_1(n_0+1)} \setminus \{0\})/\mathbb{R}_{>0} \sqcup \{[0]\}\right) \times \mathbb{R}^{n_2(n_1+1)} \cong \left(S^{n_1(n_0+1)-1} \sqcup \{*\}\right) \times \mathbb{R}^{n_2(n_1+1)}.$$

*In particular, the path–component $\widetilde{F}^\times \cong S^{n_1(n_0+1)-1} \times \mathbb{R}^{n_2(n_1+1)}$ corresponding to $v \neq 0$ has*

$$\pi_1(\widetilde{F}^\times) \cong \pi_1\left(S^{n_1(n_0+1)-1}\right) = \begin{cases} \mathbb{Z}, & n_1(n_0 + 1) = 2, \\ 0, & \textit{otherwise.} \end{cases}$$

*Thus nontrivial loops appear iff $n_1(n_0 + 1) = 2$.*

*Proof.* Write $F = \mathbb{R}^k \times \mathbb{R}^m$ with $k = n_1(n_0 + 1)$ and $m = n_2(n_1 + 1)$, and denote the blocks by $(v, u)$. The $\mathbb{R}_{>0}$–action rescales only $v$, so each orbit is of the form $\{(\lambda v, u) : \lambda > 0\}$ when $v \neq 0$, and $\{(0, u)\}$ when $v = 0$. Therefore the orbit space splits as a disjoint union of two pieces:

$$\widetilde{F} \cong \left((\mathbb{R}^k \setminus \{0\})/\mathbb{R}_{>0}\right) \times \mathbb{R}^m \sqcup \{[0]\} \times \mathbb{R}^m.$$

The standard map $[v] \mapsto v/\|v\|$ gives a homeomorphism $(\mathbb{R}^k \setminus \{0\})/\mathbb{R}_{>0} \cong S^{k-1}$. Hence

$$\widetilde{F} \cong \left(S^{k-1} \times \mathbb{R}^m\right) \sqcup \mathbb{R}^m.$$

The space is disconnected; its component with basepoints having $v \neq 0$ is $\widetilde{F}^\times \cong S^{k-1} \times \mathbb{R}^m$. Since $\mathbb{R}^m$ is contractible, Künneth's Theorem (Theorem C.4) yield $\pi_1(\widetilde{F}^\times) \cong \pi_1(S^{k-1})$. This fundamental group is $\mathbb{Z}$ precisely when $k - 1 = 1$, i.e. $k = 2$, and is trivial otherwise. Substituting $k = n_1(n_0 + 1)$ completes the proof. $\square$

**Proposition B.4.** *Let $p \geq 3$ and $m \geq 1$. Consider a single-layer neural network*

$$f_\theta : \mathbb{R}^p \to \mathbb{R}^m, \quad x \mapsto Wx + b,$$

*where $W \in \mathbb{R}^{m \times p}$ and $b \in \mathbb{R}^m$. Impose the constraint that each row of $W$ has unit norm, i.e. for each $1 \leq j \leq m$,*

$$\|W_{j,*}\| = 1,$$

*where $W_{j,*} \in \mathbb{R}^p$ is the $j$-th row of $W$. Denote by $\mathcal{F}_{norm}(p, m)$ the set of all such $(W, b)$ satisfying these constraints. Then $\mathcal{F}_{norm}(p, m)$ has nontrivial higher homotopy groups; in particular,*

$$\exists k \geq 2 \quad \textit{such that} \quad \pi_k\left(\mathcal{F}_{norm}(p, m)\right) \neq 0.$$

*Proof.* Under the given constraint, each row $W_{j,*}$ of $W$ is a point on the $(p-1)$-dimensional unit sphere $\mathbb{S}^{p-1} \subset \mathbb{R}^p$. Consequently, the entire matrix $W$ lies in the product of $m$ copies of $\mathbb{S}^{p-1}$ and then

$$\mathcal{F}_{\text{norm}}(p, m) \cong (\mathbb{S}^{p-1})^m \times \mathbb{R}^m.$$

Hence the parameter space is homeomorphic to this product manifold.

The factor $\mathbb{R}^m$ is contractible. However, $\mathbb{S}^{p-1}$ not contractible for $p \geq 2$. Since we have a product of spheres $(\mathbb{S}^{p-1})^m$, at least one factor $\mathbb{S}^{p-1}$ retains its nontrivial $(p-1)$-dimensional homotopy.

Moreover, since

$$\pi_k\big(X \times Y\big) \cong \pi_k(X) \times \pi_k(Y)$$

for topology spaces $X, Y$ and since $\mathbb{R}^m$ is contractible for all $k$ we have

$$\pi_k\Big((\mathbb{S}^{p-1})^m \times \mathbb{R}^m\Big) \cong \pi_k\big((\mathbb{S}^{p-1})^m\big).$$

Thus, if $(\mathbb{S}^{p-1})^m$ has a nonzero homotopy group in dimension $k \geq 2$, then so does $(\mathbb{S}^{p-1})^m \times \mathbb{R}^m$. Consequently,

$$\pi_{p-1}\big((\mathbb{S}^{p-1})^m\big) \neq 0 \quad \rightarrow \quad \pi_{p-1}\Big((\mathbb{S}^{p-1})^m \times \mathbb{R}^m\Big) \neq 0.$$

Since $p \geq 3$, we have found a higher homotopy group (with $k = p - 1 \geq 2$) that is nontrivial.

We have shown $\mathcal{F}_{\text{norm}}(p, m)$ is homeomorphic to a product of spheres and Euclidean space, $(\mathbb{S}^{p-1})^m \times \mathbb{R}^m$, which has a nonzero homotopy group in some dimension $k \geq 2$. Hence

$$\exists k \geq 2 \quad \text{such that} \quad \pi_k\big(\mathcal{F}_{\text{norm}}(p, m)\big) \neq 0,$$

proving the result. $\qquad\qquad\qquad\qquad\qquad\qquad\qquad\qquad\qquad\qquad\qquad\qquad\quad$ $\square$

**Example B.5.** *Take $p = 3$ and $m = 1$. Then*

$$\mathcal{F}_{\text{norm}}(3, 1) = \{(w, b) \in \mathbb{R}^{1 \times 3} \times \mathbb{R} \colon \|w\| = 1\} \cong \mathbb{S}^2 \times \mathbb{R}.$$

*Since $\pi_2(\mathbb{S}^2) \cong \mathbb{Z}$, it follows that*

$$\pi_2\big(\mathcal{F}_{\text{norm}}(3, 1)\big) \cong \pi_2(\mathbb{S}^2 \times \mathbb{R}) \cong \pi_2(\mathbb{S}^2) \cong \mathbb{Z} \neq 0.$$

*Thus even for a single-output linear layer in dimension 3, the norm constraint forces a nontrivial second homotopy group, obstructing any continuous contraction of all parameter configurations to a point.*

**Theorem B.6.** *Let $L \geq 1$ and widths $(n_0, n_1, \ldots, n_L)$ with $n_0 = p$ and $n_L = m$. Consider the parameter space*

$$\mathcal{F}_{\text{norm}}(n_0, \ldots, n_L) = \prod_{\ell=1}^{L} \Big\{(W^{(\ell)}, b^{(\ell)}) \colon W^{(\ell)} \in \mathbb{R}^{n_\ell \times n_{\ell-1}}, b^{(\ell)} \in \mathbb{R}^{n_\ell} \|W_{j,*}^{(\ell)}\| = 1 \text{ for all } 1 \leq j \leq n_\ell\Big\}.$$

*Then there is a canonical homeomorphism*

$$\mathcal{F}_{\text{norm}}(n_0, \ldots, n_L) \cong \Big(\prod_{\ell=1}^{L} (\mathbb{S}^{n_{\ell-1}-1})^{n_\ell}\Big) \times \mathbb{R}^{\sum_{\ell=1}^{L} n_\ell}.$$

*In particular, for any layer index $\ell$ with $n_{\ell-1} \geq 3$ and $n_\ell \geq 1$,*

$$\pi_{n_{\ell-1}-1}\big(\mathcal{F}_{\text{norm}}(n_0, \ldots, n_L)\big) \cong \big(\pi_{n_{\ell-1}-1}(\mathbb{S}^{n_{\ell-1}-1})\big)^{n_\ell} \cong \mathbb{Z}^{n_\ell} \neq 0.$$

*Consequently, there exists $k \geq 2$ such that $\pi_k\big(\mathcal{F}_{\text{norm}}(n_0, \ldots, n_L)\big) \neq 0$ whenever some $n_{\ell-1} \geq 3$.*

*Proof.* Fix $\ell \in \{1, \ldots, L\}$. The constraint $\|W_{j,*}^{(\ell)}\| = 1$ says that each row of $W^{(\ell)}$ lies on the unit sphere $\mathbb{S}^{n_{\ell-1}-1} \subset \mathbb{R}^{n_{\ell-1}}$. Hence the set of such weight matrices is naturally identified with the Cartesian product of $n_\ell$ spheres,

$$\{W^{(\ell)} \colon \|W_{j,*}^{(\ell)}\| = 1 \,\forall j\} \cong (\mathbb{S}^{n_{\ell-1}-1})^{n_\ell}.$$

Bias vectors $b^{(\ell)}$ are unconstrained, so they contribute a Euclidean factor $\mathbb{R}^{n_\ell}$. Taking the product over all layers yields the stated homeomorphism

$$\mathcal{F}_{\text{norm}}(n_0, \ldots, n_L) \cong \left( \prod_{\ell=1}^{L} \left( \mathbb{S}^{n_{\ell-1}-1} \right)^{n_\ell} \right) \times \mathbb{R}^{\sum_{\ell=1}^{L} n_\ell}.$$

For homotopy, note first that Euclidean factors are contractible and do not affect homotopy groups. For a finite product of based, path–connected spaces, we have

$$\pi_k \left( \prod_i X_i \right) \cong \prod_i \pi_k(X_i) \quad \text{for all } k \geq 1.$$

Applying this with $X_i = S^{n_{\ell-1}-1}$ (repeated $n_\ell$ times) and $k = n_{\ell-1} - 1$, and using $\pi_d(S^d) \cong \mathbb{Z}$, we get

$$\pi_{n_{\ell-1}-1} \left( \left( \mathbb{S}^{n_{\ell-1}-1} \right)^{n_\ell} \right) \cong \mathbb{Z}^{n_\ell}.$$

Thus, provided $n_{\ell-1} - 1 \geq 2$ (equivalently $n_{\ell-1} \geq 3$), a nontrivial higher homotopy group appears. Since this happens for any such layer, the full parameter space $\mathcal{F}_{\text{norm}}(n_0, \ldots, n_L)$ has some nonzero $\pi_k$ with $k \geq 2$ as soon as at least one input dimension $n_{\ell-1}$ is at least 3. $\qquad\square$

**Remark B.7.** *If instead of unit–norm rows one imposes stronger orthonormal–row constraints (Stiefel manifold $V_{n_{\ell-1}, n_\ell}$), the factors become Stiefel manifolds, which also have nontrivial higher homotopy. The above argument adapts by replacing spheres with $V_{n_{\ell-1}, n_\ell}$.*

Since the Euclidean factor is contractible, all homotopy and (co)homology statements below refer to

$$X = \prod_{\ell=1}^{L} \left( \mathbb{S}^{d_\ell} \right)^{n_\ell}.$$

**Corollary B.8.** *For each layer index $\ell$ with $d_\ell \geq 2$ and $n_\ell \geq 1$,*

$$\pi_{d_\ell}(X) \cong \mathbb{Z}^{n_\ell}, \quad H_{d_\ell}(X; \mathbb{Z}) \cong \mathbb{Z}^{n_\ell}, \quad \beta_{d_\ell}(X) \geq n_\ell.$$

*More generally, for any field $\mathbb{K}$, the Künneth isomorphism gives*

$$H_k(X; \mathbb{K}) \cong \bigoplus_{\substack{(i_1, \ldots, i_L) \\ \sum_j \sum_{s=1}^{n_j} \varepsilon_{j,s} d_j = k}} \bigotimes_{j=1}^{L} \bigotimes_{s=1}^{n_j} H_{\varepsilon_{j,s} d_j} \left( \mathbb{S}^{d_j}; \mathbb{K} \right), \quad \varepsilon_{j,s} \in \{0, 1\}.$$

*Proof.* Each factor $\mathbb{S}^{d_\ell}$ contributes a $\mathbb{Z}$ in degree $d_\ell$; products preserve homotopy in degree $d_\ell$ and homology via Künneth theorem. $\qquad\square$

**Corollary B.9.** *Let $r = \sum_{\ell=1}^{L} n_\ell \mathbf{1}_{\{d_\ell = 1\}}$ be the number of $\mathbb{S}^1$ factors. Then*

$$\pi_1 \left( \mathcal{F}_{\text{norm}}(n_0, \ldots, n_L) \right) \cong \mathbb{Z}^r.$$

*In particular, if some $n_{\ell-1} = 2$, the parameter space has nontrivial $\pi_1$; if all $n_{\ell-1} \geq 3$, then $\pi_1 = 0$.*

*Proof.* Note that $\pi_1$ of a finite product is the product of the $\pi_1$'s of the factors; $\pi_1(\mathbb{S}^d) = 0$ for $d \geq 2$ and $\pi_1(\mathbb{S}^1) = \mathbb{Z}$. $\qquad\square$

**Corollary B.10.** *Set*

$$k_{\min} = \min\{d_\ell : d_\ell \geq 2, n_\ell \geq 1\}.$$

*If the set is nonempty, then*

$$\pi_{k_{\min}} \left( \mathcal{F}_{\text{norm}}(n_0, \ldots, n_L) \right) \neq 0, \quad \text{and} \quad \text{rank}(\pi_{k_{\min}}) \geq \sum_{\ell: \, d_\ell = k_{\min}} n_\ell.$$

**Corollary B.11.** *If some $d_\ell \geq 2$, then $H_{d_\ell}(X; \mathbb{Z}) \neq 0$. Hence there exist nontrivial $d_\ell$–cycles in the parameter manifold. Any continuous path staying inside a sufficiently small loss sublevel neighborhood that separates these cycles cannot be contracted without exiting that neighborhood; consequently, mode connectivity within the constraint can be obstructed by these essential cycles.*

**Example B.12.** *Consider a one-hidden-layer network*

$$f_\theta(x) = W^{(2)}\sigma\big(W^{(1)}x + b^{(1)}\big) + b^{(2)},$$

*with*

$$W^{(1)} \in \mathbb{R}^{n_1 \times p}, \quad b^{(1)} \in \mathbb{R}^{n_1}, \quad W^{(2)} \in \mathbb{R}^{m \times n_1}, \quad b^{(2)} \in \mathbb{R}^m.$$

*Let $\pi$ be any permutation in the symmetric group $S_{n_1}$. Define the permutation matrix $P_\pi \in \mathbb{R}^{n_1 \times n_1}$ by*

$$(P_\pi)_{i,j} = \begin{cases} 1 & \text{if } i = \pi(j), \\ 0 & \text{otherwise.} \end{cases}$$

*Then setting*

$$W^{(1)\prime} = P_\pi W^{(1)}, \quad b^{(1)\prime} = P_\pi b^{(1)}, \quad W^{(2)\prime} = W^{(2)} P_\pi^{-1}, \quad b^{(2)\prime} = b^{(2)},$$

*we obtain a new parameter $\theta' = (W^{(1)\prime}, b^{(1)\prime}, W^{(2)\prime}, b^{(2)\prime})$ which induces the same function*

$$f_{\theta'}(x) = W^{(2)} P_\pi^{-1}\sigma\big(P_\pi(W^{(1)}x + b^{(1)})\big) + b^{(2)} = f_\theta(x).$$

*Hence each hidden-neuron permutation $\pi \in S_{n_1}$ yields an equivalence*

$$\theta \sim \theta',$$

*and the quotient of the raw parameter space by this $S_{n_1}$-action identifies all such permuted networks. This symmetry demonstrates how nontrivial topology arises even though the unconstrained space $\mathbb{R}^d$ is contractible.*

**Theorem B.13.**      *1. Each sublevel set $S_c$ is a compact, semi-algebraic subset of $\mathbb{R}^D$, and hence has finite Betti numbers*

$$\beta_k(S_c) < \infty \quad \text{(for each } k \geq 0\text{).}$$

    *2. The quotient space $S_c/G$ is also compact and semi-algebraic, so its Betti numbers $\beta_k(S_c/G)$ are finite.*

    *3. If $\beta_k(S_c/G) > 0$ for some $k \geq 1$, then there exists a nontrivial class $0 \neq [z] \in H_k(S_c/G; \mathbb{Q})$. Let $\mathrm{tr}\colon H_k(S_c/G; \mathbb{Q}) \to H_k(S_c; \mathbb{Q})$ be the transfer map (see Proposition A.3). Then*

$$\widetilde{z}\colon = \mathrm{tr}([z]) \in H_k(S_c; \mathbb{Q})^G \quad \text{satisfies} \quad \pi_*(\widetilde{z}) = [z] \neq 0.$$

*In particular $\widetilde{z} \neq 0$, so any singular $k$–cycle representing $\widetilde{z}$ is noncontractible in $S_c$.*

*Proof.*      1. First, by assumption $L$ is continuous and semi-algebraic, and $S_c = L^{-1}((-\infty, c])$ is its sublevel set. Since $L$ is semi-algebraic, the set

$$S_c = \{\theta \in \mathbb{R}^D \colon L(\theta) \leq c\}$$

is a semi-algebraic subset of $\mathbb{R}^D$ by definition it is cut out by the polynomial inequalities obtained from the piecewise-linear expressions composing $L$.

Is easy to see that $S_c$ is a closed set, and by assumption (A1) there is $R > 0$ with $S_c \subset \{\|\theta\| \leq R\}$, so $S_c$ is bounded. Hence $S_c$ is compact. Next every compact semi-algebraic set admits a triangulation into finitely many simplices (Hardt (1980)). Concretely, there exists a finite simplicial complex $K$ and a semi-algebraic homeomorphism

$$\varphi\colon |K| \to S_c,$$

where $|K|$ is the underlying polyhedron. Since $|K|$ has the homotopy type of a finite CW complex it follows that its singular homology groups $H_k(|K|; \mathbb{Q})$ are finite-dimensional vector spaces over $\mathbb{Q}$. Homology is preserved by homeomorphism, so

$$\beta_k(S_c) = \dim_\mathbb{Q} H_k(S_c; \mathbb{Q}) = \dim_\mathbb{Q} H_k(|K|; \mathbb{Q}) < \infty \quad \forall k \geq 0.$$

2. The finite group $G$ acts on $\mathbb{R}^D$ by homeomorphisms preserving $L$. In particular $G$ acts continuously on the closed, bounded set $S_c$. Since $G$ is finite, its action is proper and each orbit is finite. The quotient map

$$\pi \colon S_c \to S_c/G$$

   is surjective, closed, and open, and the quotient is Hausdorff. Moreover $\pi$ identifies finitely many points in each fiber, so $S_c/G$ inherits compactness from $S_c$. Because both $L$ and the $G$-action are semi-algebraic, the graph of the action

$$\{(\theta, g \cdot \theta) : \theta \in S_c, g \in G\}$$

   is semi-algebraic in $\mathbb{R}^D \times \mathbb{R}^D$. By Tarski–Seidenberg Theorem (Tarski (1951); Seidenberg (1954)), the image of a semi-algebraic set under a projection or quotient by a finite group is semi-algebraic. Therefore $S_c/G$ is a compact semi-algebraic subset of some Euclidean space $\mathbb{R}^N$. Applying triangulation again gives a finite simplicial complex whose underlying space is homeomorphic to $S_c/G$, and thus

$$\beta_k(S_c/G) = \dim_{\mathbb{Q}} H_k(S_c/G; \mathbb{Q}) < \infty \quad \forall k \geq 0.$$

3. Throughout we work with singular chains over $\mathbb{Q}$.

   Let $S_c = L^{-1}((-\infty, c])$ and $\pi \colon S_c \to S_c/G$ be the quotient. Since $S_c$ is compact semialgebraic and the $G$–action is semialgebraic, there exists a $G$–equivariant semialgebraic triangulation of $S_c$: a finite simplicial complex $K$ with a simplicial $G$–action and a $G$–equivariant homeomorphism $|K| \to S_c$ whose quotient $|K|/G \to S_c/G$ is a triangulation of the quotient. We identify $C_*(S_c; \mathbb{Q}) \cong C_*(K; \mathbb{Q})$ and $C_*(S_c/G; \mathbb{Q}) \cong C_*(K/G; \mathbb{Q})$.

   The simplicial $G$–action yields a right $G$–module structure on $C_*(K; \mathbb{Q})$. Passing to coinvariants gives a natural isomorphism

$$C_*(S_c/G; \mathbb{Q}) \cong C_*(S_c; \mathbb{Q})_G \tag{3}$$

   obtained by sending a simplex in the quotient to any lift and modding out by relations $g \cdot c - c$.

   Since $|G|$ is invertible in $\mathbb{Q}$, the *Reynolds operator*

$$\mathrm{Av} \colon C_*(S_c; \mathbb{Q}) \longrightarrow C_*(S_c; \mathbb{Q}), \qquad \mathrm{Av}(c) := \frac{1}{|G|} \sum_{g \in G} g_{\#} c,$$

   is a $\mathbb{Q}$–linear idempotent chain map projecting onto the $G$–invariants $C_*(S_c; \mathbb{Q})^G$. Moreover, $\mathrm{Av}$ factors through the coinvariants: if $[c] \in C_*(S_c; \mathbb{Q})_G$, define

$$s([c]) := \mathrm{Av}(c) \in C_*(S_c; \mathbb{Q}).$$

   This is well defined because $[g_{\#} c] = [c]$ in coinvariants, hence $\mathrm{Av}(g_{\#} c) = \mathrm{Av}(c)$. Composing equation 3 with $s$ yields a chain map

$$\mathrm{tr} \colon C_*(S_c/G; \mathbb{Q}) \xrightarrow{\cong} C_*(S_c; \mathbb{Q})_G \xrightarrow{s} C_*(S_c; \mathbb{Q}),$$

   the transfer at the chain level. It satisfies two identities:

$$\pi_{\#} \circ \mathrm{tr} = \mathrm{id}_{C_*(S_c/G; \mathbb{Q})} \quad \text{and} \quad \mathrm{tr} \circ \pi_{\#} = \mathrm{Av}.$$

   Indeed, the first holds because $\pi_{\#} \circ s$ is the quotient projection $C_*(S_c; \mathbb{Q}) \to C_*(S_c; \mathbb{Q})_G$ followed by the inverse of equation 3; the second holds by definition of $s$.

   Passing to homology gives natural maps

$$\mathrm{tr}_* \colon H_*(S_c/G; \mathbb{Q}) \longrightarrow H_*(S_c; \mathbb{Q}), \qquad \pi_* \colon H_*(S_c; \mathbb{Q}) \longrightarrow H_*(S_c/G; \mathbb{Q})$$

   with

$$\pi_* \circ \mathrm{tr}_* = \mathrm{id}_{H_*(S_c/G; \mathbb{Q})} \quad \text{and} \quad \mathrm{tr}_* \circ \pi_* = (\mathrm{Av})_* \text{ on } H_*(S_c; \mathbb{Q}). \tag{4}$$

   In particular, $\mathrm{tr}_*$ is injective and identifies $H_*(S_c/G; \mathbb{Q})$ with the $G$–invariant summand $H_*(S_c; \mathbb{Q})^G$, with inverse $\pi_* \vert_{H_*(S_c; \mathbb{Q})^G}$. This yields the canonical isomorphism

$$H_k(S_c/G; \mathbb{Q}) \cong H_k(S_c; \mathbb{Q})^G.$$

Now assume $\beta_k(S_c/G) > 0$. Pick any nonzero class $0 \neq [z] \in H_k(S_c/G; \mathbb{Q})$ and set

$$\widetilde{z} := \mathrm{tr}_*([z]) \in H_k(S_c; \mathbb{Q})^G.$$

By equation 4, $\pi_*(\widetilde{z}) = \pi_*\mathrm{tr}_*([z]) = [z] \neq 0$, hence $\widetilde{z} \neq 0$. Any singular $k$–cycle representing $\widetilde{z}$ is therefore nonbounding in $S_c$, i.e. noncontractible within $S_c$.

Assume $X$ is a compact $C^1$ definable $D$–manifold and $k = D - 1$.

Let $\widetilde{z} \in H_{D-1}(S_c; \mathbb{Q})$ be the nonzero invariant lift from part A and choose a representative singular cycle whose geometric support $|\widetilde{z}| \subset S_c \subset X$ is compact. By semialgebraic approximation/general position, we may assume $|\widetilde{z}|$ is a compact polyhedral $(D-1)$–subset of $X$. Alexander duality on the compact $D$–manifold $X$ gives

$$\widetilde{H}^0\big(X \setminus |\widetilde{z}|; \mathbb{Q}\big) \cong \widetilde{H}_{D-1}\big(|\widetilde{z}|; \mathbb{Q}\big).$$

The right-hand side is nonzero because $[\widetilde{z}] \neq 0$ in $H_{D-1}(X; \mathbb{Q})$ and the inclusion $H_{D-1}(|\widetilde{z}|; \mathbb{Q}) \to H_{D-1}(X; \mathbb{Q})$ maps the fundamental class of the cycle to $[\widetilde{z}]$. Hence $\widetilde{H}^0(X \setminus |\widetilde{z}|; \mathbb{Q}) \neq 0$, so $X \setminus |\widetilde{z}|$ has at least two path components.

Let $U$ be any sufficiently small open neighborhood of $|\widetilde{z}|$ in $X$. The inclusion of pairs $(X \setminus |\widetilde{z}|, \emptyset) \hookrightarrow (X \setminus U, \emptyset)$ induces an isomorphism on $\widetilde{H}^0$ for small $U$ because $U$ deformation retracts onto $|\widetilde{z}|$, so the number of path components of the complement is preserved. Thus $X \setminus U$ has at least two path components, and any path in $X$ connecting points in different components must intersect $U$.

Write $M := X/G$. In general $M$ may have orbifold singularities if the action is not free. If the support $|\widetilde{z}|$ lies in the *free locus* $X^{\mathrm{free}} := \{x \in X : G_x = \{e\}\}$, then for a sufficiently small $G$–invariant neighborhood $U$ of $|\widetilde{z}|$ the restriction $\pi \colon U \to \pi(U)$ is a covering map and $\pi(U)$ is an open subset of the $D$–manifold $M^{\mathrm{free}} := X^{\mathrm{free}}/G \subset M$. The image cycle $z := \pi_\#(\widetilde{z})$ represents the nonzero class $[z] \neq 0$ in $H_{D-1}(S_c/G; \mathbb{Q})$ by equation 4. Repeating the Alexander duality argument inside the manifold $\pi(U)$ shows that $\pi(U) \setminus |z|$ has at least two path components; hence any path in $M$ that connects points lying in distinct components of $M \setminus \pi(U)$ must intersect $\pi(U)$.

$\square$

**Example B.14.** *We compare two trained models $\theta^A$ and $\theta^B$ of identical architecture. Let layers be indexed by $\ell = 1, \ldots, L$ and let $C_\ell$ denote the number of output channels at layer $\ell$.*

*For each layer $\ell$, we estimate an affinity matrix $A_\ell \in \mathbb{R}^{C_\ell \times C_\ell}$ from a small calibration set $\mathcal{D}_{\mathrm{cal}} = \{x_m\}_{m=1}^M$ (not used for training). Let $F_{\ell,i}^A(x) \in \mathbb{R}^{H_\ell \times W_\ell}$ and $F_{\ell,j}^B(x)$ be the feature maps (post-activation) of channel $i$ and $j$ at layer $\ell$. We define pooled activations*

$$\phi(F) = \mathrm{GAP}(F) \in \mathbb{R}, \qquad \textit{(global average pooling)}$$

*and set the cosine-affinity*

$$A_\ell(i,j) = \frac{1}{M} \sum_{m=1}^M \frac{\phi\big(F_{\ell,i}^A(x_m)\big) \, \phi\big(F_{\ell,j}^B(x_m)\big)}{\|\phi(F_{\ell,i}^A)\|_2 \, \|\phi(F_{\ell,j}^B)\|_2}.$$

*We then solve a maximum-weight matching*

$$P_\ell \in \arg \max_{P \in \Pi(C_\ell)} \mathrm{trace}\big(P^\top A_\ell\big),$$

*where $\Pi(C_\ell)$ is the set of $C_\ell \times C_\ell$ permutation matrices (we use the Hungarian algorithm on the cost $C_\ell = -A_\ell$ with deterministic tie-breaks).*

*Let $W_\ell$ be the weight tensor of layer $\ell$ and $\mathrm{BN}_\ell = (\gamma_\ell, \beta_\ell, \mu_\ell, \sigma_\ell^2)$ its batch-norm parameters. We define the aligned copy of $\theta^B$ by applying $P_\ell$ to* output *channels of layer $\ell$ and the inverse permutation to the corresponding* input *channels of the next layer:*

$$\widehat{W}_\ell^B = P_\ell W_\ell^B P_{\ell-1}^\top, \quad \widehat{\gamma}_\ell^B = P_\ell \gamma_\ell^B, \quad \widehat{\beta}_\ell^B = P_\ell \beta_\ell^B, \quad \widehat{\mu}_\ell^B = P_\ell \mu_\ell^B, \quad \widehat{\sigma}_\ell^{2,B} = P_\ell \sigma_\ell^{2,B},$$

*with $P_0 = I$ for the input layer.*

---

**Algorithm 1** Permutation matching and BN-aware interpolation

---

**Input:** trained models $\theta^A, \theta^B$; calibration set $\mathcal{D}_{\text{cal}}$; grid $\mathcal{A} \subset [0,1]$
**Output:** loss/accuracy along $\alpha \in \mathcal{A}$

1. For each layer $\ell$: compute $A_\ell$ on $\mathcal{D}_{\text{cal}}$, solve $P_\ell = \text{Hungarian}(-A_\ell)$.

2. Form aligned $\widehat{\theta}^B$ by permuting channels with $\{P_\ell\}$ as above.

3. For each $\alpha \in \mathcal{A}$:

   (a) Set $\theta_\alpha = (1-\alpha)\theta^A + \alpha\,\widehat{\theta}^B$.
   (b) *BN recalibration:* enable BN stat updates, disable dropout; run $\theta_\alpha$ on $\mathcal{D}_{\text{cal}}$ (no gradients) to refresh $\{\mu_\ell, \sigma_\ell^2\}$.
   (c) Evaluate $\mathcal{L}(\theta_\alpha)$ and accuracy on $\mathcal{D}_{\text{val}}$ in eval mode.

---

| $\alpha$ | $\mathcal{L}_{\text{raw}}$ | $\mathcal{L}_{\text{aln}}$ | $\mathcal{L}_{\text{aln+BN}}$ | $\text{Acc}_{\text{raw}}$ | $\text{Acc}_{\text{aln}}$ | $\text{Acc}_{\text{aln+BN}}$ |
|---|---|---|---|---|---|---|
| 0.00 | 2.3068 | 2.3068 | 2.3060 | 0.0938 | 0.0938 | 0.0938 |
| 0.10 | 2.3067 | 2.3067 | 2.3060 | 0.0938 | 0.0938 | 0.0938 |
| 0.20 | 2.3066 | 2.3067 | 2.3059 | 0.0938 | 0.0938 | 0.0938 |
| 0.30 | 2.3065 | 2.3066 | 2.3059 | 0.0938 | 0.0938 | 0.0938 |
| 0.40 | 2.3064 | 2.3066 | 2.3060 | 0.0938 | 0.0938 | 0.0938 |
| 0.50 | 2.3063 | 2.3065 | 2.3060 | 0.0938 | 0.0938 | 0.0938 |
| 0.60 | 2.3063 | 2.3065 | 2.3060 | 0.0938 | 0.0938 | 0.0938 |
| 0.70 | 2.3063 | 2.3066 | 2.3061 | 0.0938 | 0.0938 | 0.0938 |
| 0.80 | 2.3063 | 2.3066 | 2.3061 | 0.0938 | 0.0938 | 0.0938 |
| 0.90 | 2.3063 | 2.3067 | 2.3062 | 0.0938 | 0.0938 | 0.0938 |
| 1.00 | 2.3063 | 2.3067 | 2.3062 | 0.0938 | 0.0938 | 0.0938 |

Table 1: Interpolation losses and accuracies along $\alpha$ with and without alignment and BN recalibration.

*We linearly interpolate in the aligned parameter space*

$$\theta_\alpha \;=\; (1-\alpha)\,\theta^A \;+\; \alpha\,\widehat{\theta}^B, \qquad \alpha \in [0,1].$$

*This path remains within a single path-component of the quotient space $S_c/G$ whenever $\theta^A$ and $\widehat{\theta}^B$ lie in the same sublevel component.*

*To avoid spurious barriers from stale running statistics, we recompute batch-norm running means/variances at each $\alpha$ using $\mathcal{D}_{\text{cal}}$, without updating weights. Also we use $M \in [1k, 5k]$ calibration examples, BN momentum $m \in [0.01, 0.1]$, and $T = 1$ epoch over $\mathcal{D}_{\text{cal}}$ unless stated otherwise.*

**Corollary B.15.** *Let $X = S_c/G$ and suppose $\beta_k(X) > 0$ for some $k > 0$. Then there exists a nonzero class $[z] \in H_k(X; \mathbb{Q})$. Fix a singular $k$–cycle $z$ representing $[z]$, let $|z| \subset X$ be the union of the images of the simplices appearing in $z$, and let $U$ be a sufficiently small open neighborhood of $|z|$ in $X$.*

*Assume there are two critical points $\theta^A, \theta^B \in S_c$ such that their images $[\theta^A], [\theta^B] \in X$ lie in different path components of $X \setminus U$. Then:*

1. *There is no continuous path in $X$ connecting $[\theta^A]$ to $[\theta^B]$ while staying inside $X \setminus U$; hence every path in $X$ between them must intersect $U$.*

2. *In particular, since $|z| \subset L^{-1}(c)/G \subset X$, every path in the* open *sublevel set $\{L < c\}/G$ connecting $[\theta^A]$ to $[\theta^B]$ is impossible; equivalently, any path in parameter space that connects $\theta^A$ to $\theta^B$ must leave $\{L < c\}$ and hit the level set $L^{-1}(c)$.*

*Proof.* Recall that a singular $k$–cycle on $X$ is a finite $\mathbb{Q}$–linear combination $z = \sum_i a_i \sigma_i$ of singular $k$–simplices $\sigma_i : \Delta^k \to X$ with $\partial z = 0$. Its homology class $[z] \in H_k(X; \mathbb{Q})$ is nonzero iff $z$ is not a boundary.

Since $\beta_k(X) > 0$, choose a nonzero class $[z] \in H_k(X; \mathbb{Q})$ and a cycle representative $z$ as above. Let $|z|$ be the geometric support of $z$, and $U$ a sufficiently small open neighborhood of $|z|$ so that $U$ deformation–retracts onto $|z|$. By hypothesis, $[\theta^A]$ and $[\theta^B]$ lie in different path components of $X \setminus U$. Thus no continuous path contained in $X \setminus U$ can connect them; hence any path in $X$ must meet $U$.

Now, because $X$ is the closed sublevel quotient, and the cycle support $|z|$ lies in the boundary level $\left(L^{-1}(c)\right)/G \subset X$, every neighborhood $U$ of $|z|$ is contained in $X$ but intersects the boundary level. Therefore any path in the open sublevel set $\{L < c\}/G$ cannot intersect $U$, so by (1) such a path cannot exist. Equivalently, any path in parameter space connecting $\theta^A$ to $\theta^B$ must exit $\{L < c\}$ and hit the level set $L^{-1}(c)$. □

**Example B.16.** *Let $p = 2$, $m = 1$, and consider data drawn on the unit circle $\mathbb{S}^1 \subset \mathbb{R}^2$ labeled by the sign of the first coordinate $y = \text{sign}(x_1)$. Take the network $f_\theta(x) = w_2^\top \text{ReLU}(W_1 x + b_1) + b_2$, and define the hinge loss $L(\theta) = \frac{1}{N} \sum_{i=1}^{N} \max\{0, 1 - y_i f_\theta(x_i)\}$. For a moderate threshold $c \approx 0.1$, the sublevel set $S_c/G$ inherits, after quotienting by the permutation-and-scaling group $G$, precisely a deformation retract onto a copy of $\mathbb{S}^1$ (the set of separating hyperplanes that cut the circle into two arcs). We check that $\beta_1(S_c/G) = 1$, so there is a one-dimensional loop of equivalent low-loss classifiers. Any two minimizers lying on distinct sides of this loop require a loss barrier to connect.*

*In Figure 7 with a very small scale shows only short $H_0$ bars near the diagonal and no persistent $H_1$ feature. This corresponds to the un–canonicalized cloud, which is essentially a contractible cluster; there is no loop once gauge redundancy is ignored, hence $\beta_1 \approx 0$ at this scale.*

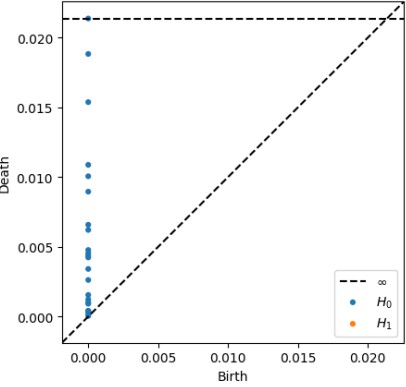 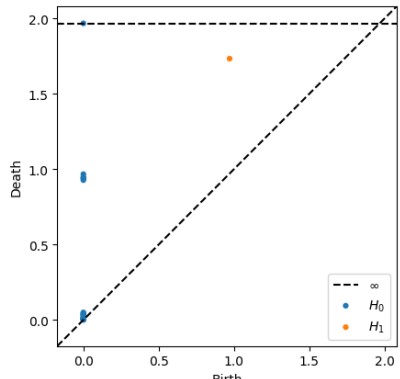

Figure 7: $\beta_1 = 0$            Figure 8: $\beta_1 = 1$

*At a larger filtration scale the diagram (Figure 8) displays a single long–lived $H_1$ class far from the diagonal together with one connected component in $H_0$. This is the hallmark of a circle: $\beta_0 = 1$, $\beta_1 = 1$, higher Betti numbers 0. It confirms that, after quotienting by $G$ (or after rotate–back), the set of low–loss separators forms a 1–cycle.*

**Remark B.17.** *If for some $k$ the $(k-1)$th Betti number satisfies $\beta_{k-1}(S_c/G) = B > 0$, then the sublevel set $S_c/G$ must have at least $B + 1$ path-connected components in its complement of a suitable $(k-1)$-cycle. In particular, there are at least $B + 1$ distinct basins of attraction for gradient descent at level c.*

*Embed $S_c/G$ as a compact subset of the sphere $\mathbb{S}^N$ by one-point compactification in $\mathbb{R}^N$. Denote the embedded copy by $A \subset \mathbb{S}^N$. By Theorem Alexander duality (Hatcher (2000)),*

$$\widetilde{H}^{N-(k-1)-1}\left(\mathbb{S}^N \setminus A; \mathbb{Q}\right) \cong \widetilde{H}_{k-1}(A; \mathbb{Q}) \cong \mathbb{Q}^B.$$

*Hence $H^{N-k}\left(\mathbb{S}^N \setminus A; \mathbb{Q}\right)$ has dimension B. Since cohomology $H^0$ counts connected components, for $k = 1$ this says $H^0(\mathbb{S}^N \setminus A; \mathbb{Q}) \cong \mathbb{Q}^B$, so $\mathbb{S}^N \setminus A$ has B path-components. More generally, non-vanishing $H^{N-k}$ implies that $\mathbb{S}^N \setminus A$ cannot be covered by fewer than $B + 1$ open sets each*

*trivial in cohomology; in practical terms, there must be at least $B + 1$ connected regions in the complement of any representative $(k - 1)$-cycle in $A$. Translating back to $S_c/G$, this means that if one removes a single nontrivial $(k - 1)$-cycle from $S_c/G$, the remaining set splits into at least $B + 1$ path-components. Each component is an open region in which one may place distinct initializations for gradient-based training. Since continuous deformation between components would cross the removed cycle, these yield $B + 1$ distinct basins of low-loss attraction.*

Let $L : \mathbb{R}^D \to \mathbb{R}$ be a $C^2$ semi-algebraic function. A point $\theta^*$ is a *Morse critical point* if $\nabla L(\theta^*) = 0$ and $\det \nabla^2 L(\theta^*) \neq 0$. Its *Morse index* is the number of negative eigenvalues of the Hessian.

**Theorem B.18** (Morse Lemma). *In local coordinates near a nondegenerate critical point of index $\lambda$, we have*

$$L(x) = L(\theta^*) - (x_1^2 + \cdots + x_\lambda^2) + (x_{\lambda+1}^2 + \cdots + x_D^2).$$

As a consequence, sublevel sets change topology only at critical values via cell attachments. These changes are reflected in homotopy groups $\pi_k$ and cohomology groups $H^k$.

**Theorem B.19.** *Let $\mathcal{F} \cong \mathbb{R}^D$ be the parameter space of a finite-dimensional neural network with continuous, piecewise-linear activation functions, and let*

$$L \colon \mathcal{F} \to \mathbb{R}$$

*be a smooth semi-algebraic loss function. Assume the following.*

- *(M) $L$ is a Morse function on $\mathcal{F}$, i.e., all critical points $\theta^*$ satisfy $\det\big(\nabla^2 L(\theta^*)\big) \neq 0$.*

- *(G) The gradient-descent trajectory*

$$\gamma \colon [0, 1] \to \mathcal{F}, \quad \dot\gamma(t) = -\nabla L\big(\gamma(t)\big),$$

  *is transverse to all level sets of $L$ and avoids degenerate encounters with critical values.*

*Fix $\varepsilon, \delta > 0$ sufficiently small, and for each $t \in [0, 1]$ define the local sublevel neighborhood*

$$N_t = \big\{ \theta \in \mathcal{F} \colon \|\theta - \gamma(t)\| \leq \varepsilon, \, L(\theta) \leq L\big(\gamma(t)\big) + \delta \big\}.$$

*Then there exist finitely many bifurcation times*

$$0 < t_1 < \cdots < t_k < 1,$$

*namely the times at which $\gamma(t_i)$ is a critical point of $L$, with the following property.*

- *For each open interval $I \subset [0, 1] \setminus \{t_1, \ldots, t_k\}$, the inclusion maps*

$$N_{t'} \hookrightarrow N_{t''} \quad (t', t'' \in I, t' < t'')$$

  *are all homotopy equivalences.*

- *At each $t_i$, the homotopy type of $N_{t_i - \epsilon}$ changes by the attachment of a single cell of dimension equal to the Morse index of the critical point $\gamma(t_i)$.*

*In particular, the Betti numbers of $N_t$ jump exactly at the $t_i$, and these jumps correspond to observable phase transitions in the training dynamics.*

*Proof.* Since $L$ is a Morse function, its critical values $c_1 < c_2 < \cdots < c_k$ are finitely many and isolated. By Assumption G, the composite $t \mapsto L(\gamma(t))$ meets each $c_i$ exactly once, defining unique times $t_i \in (0, 1)$.

Now fix an open interval $I \subset [0, 1] \setminus \{t_1, \ldots, t_k\}$. For any $t \in I$, $\gamma(t)$ lies on a regular level set $L^{-1}(L(\gamma(t)))$. Since $\|\nabla L\| > 0$ there, the Implicit Function Theorem gives a neighborhood $U_t$ on which the sublevel sets

$$U_t \cap L^{-1}((-\infty, L(\gamma(t)) + \delta])$$

are $C^1$-diffeomorphic as $t$ varies in a small subinterval. Moreover, the gradient flow of $-\nabla L$ provides an explicit deformation retraction from $N_{t'}$ onto $N_{t''}$ for any $t' < t''$ in $I$, by following flow lines until reaching the level $L(\gamma(t''))$. This establishes that the inclusion $N_{t'} \hookrightarrow N_{t''}$ is a homotopy equivalence.

Fixing $i$ and setting $\theta_i^* = \gamma(t_i)$, $c = c_i = L(\theta_i^*)$ we have that there exist local coordinates $(x_1, \ldots, x_D)$ on $B_i = \{\|\theta - \theta_i^*\| \leq \varepsilon\}$ such that

$$L(x) = c - (x_1^2 + \cdots + x_{\lambda_i}^2) + (x_{\lambda_i+1}^2 + \cdots + x_D^2),$$

where $\lambda_i$ is the index of the Hessian $\nabla^2 L(\theta_i^*)$. Define in these coordinates the two sublevel sets inside $B_i$:

$$A_- = \{\|x\| \leq \varepsilon, L(x) \leq c - \eta\},$$
$$A_+ = \{\|x\| \leq \varepsilon, L(x) \leq c + \eta\},$$

where $0 < \eta < \delta$ is chosen so that $c \pm \eta$ are regular values and the region between levels does not contain other critical points. Is easy to see that $A_+$ is homeomorphic to $A_-$ with a $\lambda_i$–cell attached along its boundary sphere $\mathbb{S}^{\lambda_i-1}$. Concretely,

$$A_+ \cong A_- \cup \left(D^{\lambda_i} \times \{0\}\right),$$

where $D^{\lambda_i} = \{(x_1, \ldots, x_{\lambda_i}): x_1^2 + \cdots + x_{\lambda_i}^2 \leq \eta\}$.

Choose $\eta$ small enough that $N_{t_i-\epsilon} \cap B_i = A_-$ and $N_{t_i+\epsilon} \cap B_i = A_+$ for sufficiently small $\epsilon > 0$. Outside $B_i$, $\gamma(t)$ remains on regular levels, so by Step 2 the inclusion of the complements is a homotopy equivalence. Using Mayer–Vietoris Theorem we can glue the cell-attachment in $B_i$ to the global $N_t$, concluding that passing from $t_i - \epsilon$ to $t_i + \epsilon$ attaches exactly one $\lambda_i$–cell to $N_{t_i-\epsilon}$, yielding $N_{t_i+\epsilon}$.

Since there are only $k$ critical times, the homotopy type of $N_t$ changes only at the finitely many $t_i$. Each cell-attachment in dimension $\lambda_i$ increases the Betti number $\beta_{\lambda_i}$ by one and does not affect $\beta_j$ for $j \neq \lambda_i$. Away from these times, Betti numbers remain constant by homotopy invariance. $\qquad\square$

**Example B.20.** *Define*

$$L: \mathbb{R}^2 \to \mathbb{R},$$
$$L(w_1, w_2) = \left(w_1^2 - 1\right)^2 + w_2^2.$$

*A straightforward computation shows that we have two global minima at $\theta_\pm = (\pm 1, 0)$, $L(\theta_\pm) = 0$, each nondegenerate of index 0, and one saddle at $\theta_0 = (0,0)$, $L(\theta_0) = 1$, with Hessian eigenvalues $\{-2, 2\}$, hence index $\lambda = 1$.*

*Let $\gamma(t)$ be the solution of*

$$\dot{\gamma}(t) = -\nabla L\left(\gamma(t)\right), \quad \gamma(0) = (2, 1),$$

*reparametrized so that $t \in [0,1]$ and $L(\gamma(t))$ decreases monotonically from $L(2,1) = 10$ to 0. Is easy to see that $\gamma(t)$ passes through the saddle exactly once at some $t_1 \in (0,1)$ with $\gamma(t_1) = \theta_0$.*

*Now fix small $\varepsilon, \delta > 0$ so that the closed ball $B = \{\|w - \theta_0\| \leq \varepsilon\}$ contains no critical point except $\theta_0$, and $L^{-1}(1 \pm \delta)$ are regular levels in $B$. For each $t$ define*

$$N_t = \left\{w \in \mathbb{R}^2: \|w - \gamma(t)\| \leq \varepsilon, L(w) \leq L(\gamma(t)) + \delta\right\}.$$

*See the following*

- *For $t < t_1$, $\gamma(t)$ lies on a level $L(\gamma(t)) > 1$, so $N_t$ is homeomorphic to a filled disk (contractible).*

- *At $t = t_1$, $\gamma(t_1) = \theta_0$ is the saddle of index $\lambda = 1$. By the Morse-lemma coordinates (setting $x = w_1$, $y = w_2$),*

$$L(x, y) = 1 - x^2 + y^2 \quad \text{near } (0,0),$$

  *so passing from $L \leq 1 - \delta$ to $L \leq 1 + \delta$ attaches a 1–cell. Topologically, $N_{t_1+\epsilon}$ is a disk with a one-dimensional handle through it, i.e. two disks joined by a narrow corridor.*

- *For $t > t_1$, since $L(\gamma(t)) < 1$, the neighborhood $N_t$ splits into two contractible components each containing one minimum $\theta_+$ or $\theta_-$. Thus $N_t$ has two path components.*

*Using Theorem B.19 we can see that on $(0, t_1)$ and on $(t_1, 1]$, all inclusions $N_{t'} \hookrightarrow N_{t''}$ are homotopy equivalences (each side stays either a single disk or two disks). Also, at $t_1$, $N_{t_1-\epsilon}$ changes to $N_{t_1+\epsilon}$, exactly by attaching one 1–cell, matching the saddle's index.*

**Definition B.21.** *For $d \geq 1$, the special orthogonal group*

$$\mathrm{SO}(d) = \{R \in \mathrm{GL}(d, \mathbb{R}) : R^\top R = I, \det R = +1\}$$

*is the Lie group of all orientation-preserving isometries of Euclidean $\mathbb{R}^d$.*

This group is compact, connected for $d \geq 2$, and the fundamental group is $\{0\}$ when $d = 2$ and $\mathbb{Z}/2\mathbb{Z}$ when $d \geq 3$. Its universal cover is the spin group $\mathrm{Spin}(d)$, a nontrivial two-fold cover when $d \geq 3$.

We recall models of hyperbolic space $\mathcal{H}^d$ ($d \geq 3$):

- Poincaré ball model $\{u \in \mathbb{R}^d : \|u\| < 1\}$ (see Figure 9 (a)).

- Upper-half-space $\{(x, y) \in \mathbb{R}^{d-1} \times \mathbb{R}_{>0}\}$ (see Figure 9 (b)).

- Lorentz Hyperboloid model $\{x \in \mathbb{R}^{d+1} : -x_0^2 + \sum_{i=1}^d x_i^2 = -1, x_0 > 0\}$ (see Figure 9 (c))

All those model satisfy $\mathrm{Isom}^+(\mathcal{H}^d) \cong \mathrm{SO}_0(d, 1)$ with maximal compact subgroup $\mathrm{SO}(d)$. The Iwasawa decomposition (Theorem C.2) $\mathrm{SO}_0(d, 1) \simeq SO(d) \times \mathbb{R}^d$ deformation-retracts onto $SO(d)$, so $\pi_1(SO_0(d, 1)) \cong \pi_1(SO(d)) \cong \mathbb{Z}/2\mathbb{Z}$. Here we consider $\mathbb{R}^{d+1}$ equipped with the Lorentz form of signature $(-, +, \ldots, +)$, $\langle x, y \rangle = -x_0 y_0 + x_1 y_1 + \cdots + x_d y_d$. The Lorentz group is given by $\mathcal{L}(d, 1) = \{M \in \mathrm{GL}(d+1, \mathbb{R}) : M^\top \eta M = \eta\}$, $\eta = \mathrm{diag}(-1, 1, \ldots, 1)$, and the special Lorentz group is $\mathrm{SO}(d, 1) = \{M \in \mathcal{L}(d, 1) : \det M = +1\}$. The identity component $\mathrm{SO}_0(d, 1)$ consists of those matrices preserving time-orientation.

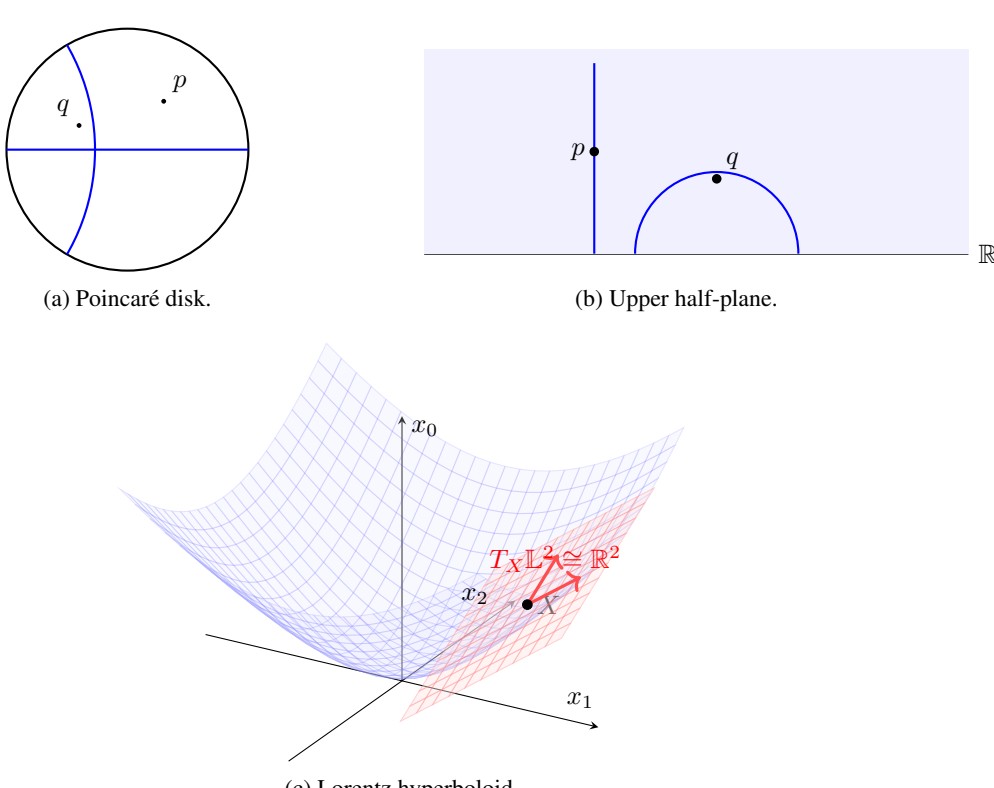

(a) Poincaré disk.

(b) Upper half-plane.

(c) Lorentz hyperboloid.

Figure 9: Three isometric hyperbolic models.

**Lemma B.22.** *Let $d \geq 3$ and let $G = \mathrm{SO}_0(d, 1)$ be the identity component of the Lorentz group, realized as the group of linear isometries of the Lorentz model of hyperbolic $d$-space. Consider a*

*hyperbolic neural network of depth L, whose parameters consist of*

$$(T^{(1)}, T^{(2)}, \ldots, T^{(L)}, b^{(1)}, \ldots, b^{(L)}),$$

*where each weight-layer transformation $T^{(\ell)} \in G$ (implementing a Möbius linear map) and each bias $b^{(\ell)} \in \mathbb{H}^d$ (the hyperboloid model). Then the parameter space*

$$\mathbb{F}_{\mathrm{hyp}} = G^L \times (\mathbb{H}^d)^L$$

*has nontrivial fundamental group and nonzero first cohomology*

$$\pi_1(\mathbb{F}_{\mathrm{hyp}}) \cong \pi_1(G) \cong \pi_1\big(\mathrm{SO}(d)\big) \cong \mathbb{Z}/2\mathbb{Z}, \quad H^1\big(\mathbb{F}_{\mathrm{hyp}}; \mathbb{Z}/2\mathbb{Z}\big) \neq 0.$$

*In particular, there exist noncontractible loops in parameter space that reflect the underlying negative-curvature geometry of the model.*

*Proof.* The identity component $G = \mathrm{SO}_0(d, 1)$ is a noncompact, connected Lie group whose maximal compact subgroup is $K = \mathrm{SO}(d)$. By the Cartan decomposition, there is a diffeomorphism (global Iwasawa decomposition)

$$G \cong K \times \mathbb{R}^d$$

as a manifold. In particular $G$ deformation-retracts onto $K$. Since deformation retraction preserves homotopy groups,

$$\pi_1(G) \cong \pi_1(K) \cong \pi_1\big(\mathrm{SO}(d)\big).$$

For $d \geq 3$, it is a classical fact that

$$\pi_1\big(\mathrm{SO}(d)\big) \cong \mathbb{Z}/2\mathbb{Z} \quad \text{(the nontrivial two-fold cover } p(d) \to \mathrm{SO}(d)).$$

Hence $G$ admits a noncontractible loop of order two.

The hyperbolic neural network parameter space factors as

$$\mathbb{F}_{\mathrm{hyp}} = G^L \times (\mathbb{H}^d)^L.$$

Each hyperbolic bias factor $\mathbb{H}^d$ is diffeomorphic to $\mathbb{R}^d$ (hence contractible), so

$$\pi_1(\mathbb{F}_{\mathrm{hyp}}) \cong \pi_1(G^L) \cong \big(\pi_1(G)\big)^L \cong (\mathbb{Z}/2\mathbb{Z})^L$$

is nontrivial as soon as $L \geq 1$. In particular, there are noncontractible loops in $\mathbb{F}_{\mathrm{hyp}}$.

Using the Künneth formula and the fact that $H^1(\mathbb{Z}/2\mathbb{Z}; \mathbb{Z}/2\mathbb{Z}) \cong \mathbb{Z}/2\mathbb{Z}$, we obtain

$$H^1\big(\mathbb{F}_{\mathrm{hyp}}; \mathbb{Z}/2\mathbb{Z}\big) \cong H^1\big(G^L; \mathbb{Z}/2\mathbb{Z}\big) \cong \big(H^1(G; \mathbb{Z}/2\mathbb{Z})\big)^L \neq 0.$$

Equivalently, the first cohomology group of the parameter space is nonzero, detecting these loops.

Thus any hyperbolic neural network parameterized via Möbius operations necessarily has a parameter manifold with nontrivial topology reflecting the negative curvature symmetry group of the model. $\square$

**Remark B.23.** *The Lemma B.22 is valid for any isometric Hyperbolic model.*

## C  AUXILIARY RESULTS

**Theorem C.1** (Mayer–Vietoris)**.** *Let $X$ be a topological space and $U, V \subset X$ be subspaces such that $X = U \cup V$. For any coefficient ring $R$ there is a natural long exact sequence in singular homology*

$$\cdots \to H_n(U \cap V; R) \xrightarrow{(i_*, j_*)} H_n(U; R) \oplus H_n(V; R) \xrightarrow{k_* - \ell_*} H_n(X; R) \xrightarrow{\partial} H_{n-1}(U \cap V; R) \to \cdots$$

*where $i\colon U \cap V \hookrightarrow U$, $j\colon U \cap V \hookrightarrow V$, $k\colon U \hookrightarrow X$, $\ell\colon V \hookrightarrow X$ are inclusions, $(i_*, j_*)$ is the direct sum of the induced maps, $k_* - \ell_*$ is their difference, and $\partial$ is the connecting homomorphism.*

*Also we have*

$$\cdots \to H^{n-1}(U \cap V; R) \xrightarrow{\delta} H^n(X; R) \xrightarrow{(k^*, \ell^*)} H^n(U; R) \oplus H^n(V; R) \xrightarrow{i^* - j^*} H^n(U \cap V; R) \to \cdots$$

*where the maps are induced by inclusions and $\delta$ is the cohomological connecting homomorphism.*

**Theorem C.2** (Iwasawa decomposition). *Let $G$ be a connected real semisimple Lie group with Lie algebra $\mathfrak{g}$. Fix a Cartan involution $\theta$ on $\mathfrak{g}$ and write the Cartan decomposition $\mathfrak{g} = \mathfrak{k} \oplus \mathfrak{p}$, where $\mathfrak{k}$ (resp. $\mathfrak{p}$) is the $+1$ (resp. $-1$) eigenspace of $\theta$. Let $\mathfrak{a} \subset \mathfrak{p}$ be a maximal abelian subspace and let $\Sigma \subset \mathfrak{a}^*$ be the corresponding (restricted) root system. Choose a set of positive roots $\Sigma^+ \subset \Sigma$ and set*

$$\mathfrak{n} = \sum_{\alpha \in \Sigma^+} \mathfrak{g}_\alpha, \quad A = \exp(\mathfrak{a}), \quad N = \exp(\mathfrak{n}),$$

*and let $K \subset G$ be the maximal compact subgroup with Lie algebra $\mathfrak{k}$. Then the multiplication map*

$$K \times A \times N \to G, \quad (k, a, n) \longmapsto kan$$

*is a diffeomorphism onto $G$. In particular, every $g \in G$ can be written uniquely as $g = kan$ with $k \in K$, $a \in A$, $n \in N$, and this decomposition depends smoothly on $g$.*

**Remark C.3.** *On the Lie algebra level one has the vector space decomposition $\mathfrak{g} = \mathfrak{k} \oplus \mathfrak{a} \oplus \mathfrak{n}$, compatible with the group-level factorization $G = KAN$. The choice of $\Sigma^+$ is not canonical, but different choices yield conjugate $AN$-subgroups.*

**Theorem C.4.** *[Künneth formula, field coefficients] Let $X, Y$ be topological spaces and $F$ a field. Then for each $n \geq 0$ there is a natural graded isomorphism*

$$H_n(X \times Y; F) \cong \bigoplus_{p+q=n} H_p(X; F) \otimes_F H_q(Y; F).$$

*Dually, there is a natural graded isomorphism*

$$H^n(X \times Y; F) \cong \bigoplus_{p+q=n} H^p(X; F) \otimes_F H^q(Y; F),$$

*compatible with the cup product and the cross product via the standard identification.*

The following Theorem provides the Alexander duality (see Hatcher (2000)).

**Theorem C.5.** *If $K$ is a compact, locally contractible subspace of a closed orientable $n$-manifold $M$, then $H_i(M, M \setminus K; \mathbb{Z}) \cong H^{n-i}(K; \mathbb{Z})$ for all $i$.*

# D    USE OF LLMS

In accordance with Policy 1, we disclose the use of an LLM as a writing and editing assistant. The LLM was used to improve grammar, wording, typography and generate auxiliary LaTeX tables.

