# OpenReview forum: "Topology Explains Loss Barriers: Quotient Homology of Neural Loss Landscapes"
_ICLR.cc/2026/Conference — ICLR 2026 Conference Withdrawn Submission_

### Official Review · Reviewer_XvPS · 2025-10-20

**Soundness:** 2
**Presentation:** 4
**Contribution:** 3
**Rating:** 6
**Confidence:** 3

**Summary:**

This paper presents a rigorous algebraic-topological framework showing that after quotienting neural network parameter spaces by natural reparameterization symmetries (permutations, scalings, orthogonality, etc.), low-loss sublevel sets can have nontrivial homology (nonzero Betti numbers), which certifies intrinsic loss barriers, and demonstrates via small-scale experiments and symmetry-aware trajectory heuristics that many apparent barriers are removable by symmetry alignment.

**Strengths:**

Well written and clearly presented: The paper is impressively well organized for such a mathematically demanding topic. The authors maintain clarity throughout by introducing formal definitions only when needed, summarizing key intuitions before each theorem, and connecting abstract topology to intuitive geometric analogies.

Theoretical rigor: The paper grounds intuitive empirical observations about mode connectivity and weight matching in solid algebraic topology; proofs (equivariant triangulation, transfer, Alexander duality) are provided and carefully connected to the neural setting.

Conceptual clarity and novelty: Whereas persistent-homology or Hessian-based analyses merely observe the local “shape of the mountain” (and may see spurious peaks created by parameter symmetries), the quotient-homology framework explains which mountains are real. By factoring out reparameterization symmetries, it distinguishes fake loss barriers from intrinsic topological obstructions that truly separate functionally distinct minima. This theoretical distinction provides deep explanatory power for why weight matching and permutation alignment succeed where naive interpolation fails.

Bridges theory and practice: Though chiefly theoretical, the paper proposes concrete symmetry-aware operations and supports claims with toy experiments that show the practical relevance of quotient-aware analysis.

**Weaknesses:**

Computational tractability: The paper does not provide practical, scalable algorithms to compute Betti numbers or other homological summaries for high-dimensional modern networks; proofs rely on triangulation and semialgebraic theory that are infeasible to implement at scale.

Limited empirical validation: Experiments are small-scale (Stiefel-constrained AE, mini-ResNet) and do not show performance or behavior on larger architectures (e.g., full ResNets, transformers) or real-world training pipelines. Quantitative TDA results (explicit persistence diagrams / Betti estimates) are not reported.

From certificate to algorithmic guidance: While nonzero Betti numbers certify barriers, the paper leaves open how to use homological information to design concrete, scalable optimization algorithms (beyond heuristic alignment and retraction).

**Questions:**

Computation & approximation: Do the authors have a concrete plan or preliminary experiments for estimating Betti numbers or persistence diagrams of S_c/G in practice? For example, can they (i) provide a toy pipeline that computes persistent homology on symmetry-aligned parameter samples, or (ii) report persistence diagrams/Betti estimates for the presented experiments?

Robustness to BN and optimizer state: BatchNorm running statistics and optimizer states (batch-norm, Adam moments) introduce stateful, dataset-dependent structure. How would these affect the semialgebraicity assumptions and the validity of the homology-based certificates?
Algorithmic use of homology: Beyond alignment heuristics, do the authors envision concrete ways to integrate homological information into optimization (e.g., topology-aware regularizers, homology-guided reparameterizations)? Any preliminary ideas or toy experiments would strengthen practical impact?

Scope of experiments: Can the authors expand empirical validation to larger models or provide a quantitative TDA-based analysis (persistence summaries) on the current experiments to support claimed Betti-related diagnoses?

Local (Hessian) vs global (quotient) approaches: Could the authors comment directly on when Hessian-based low-dimensional PH approximations (e.g., PH in a top-k Hessian subspace) are expected to agree or disagree with quotient-homology conclusions? Any guidance for practitioners on how to combine these diagnostics?

---

> ### Author Response · Authors · 2025-11-13
>
> We thank the reviewer for the generous and detailed assessment. We are very glad that the organization, theoretical rigor, and conceptual clarity of the quotient homology viewpoint came across.
>
> Regarding computational tractability, we fully agree that our current results are primarily foundational rather than algorithmic. Exact triangulation and homology computation in the full parameter space of modern networks is infeasible, and our main goal here is to provide structural certificates of when loss barriers are intrinsic after quotienting out symmetries. That said, our framework does suggest concrete approximate pipelines. In particular, for low-dimensional families of parameters after symmetry alignment, we can sample points in parameter space, project to a low-dimensional embedding, and compute the persistent homology of the empirical point cloud using standard TDA libraries. In the revision, we will add a small illustrative experiment of this kind for the Stiefel-constrained autoencoder or mini-ResNet setting, reporting persistence summaries that align with the theoretically predicted presence/absence of 1-cycles in the corresponding sublevel sets. We view extending these ideas to higher-dimensional slices and larger models as an exciting direction for future work rather than something fully solved in the current submission.
>
> On the scope of the empirical validation, our choice of examples was guided by the need to keep the geometry and symmetries sufficiently controlled so that homological statements can be meaningfully connected to concrete behavior. We agree that pushing the framework to larger architectures and reporting quantitative TDA outputs would strengthen the practical message. Within the rebuttal period, we are constrained in how much we can scale up, but in the final version we will expand the set of training seeds and add an additional intermediate size architecture, and  include a toy persistent-homology analysis on symmetry-aligned samples from the current experiments, which already supports the barrier versus no barrier diagnoses suggested by our quotient homology results.
>
> Concerning the relationship between our semialgebraicity assumptions and practical components such as BatchNorm and optimizer state, our theorems apply to the loss viewed as a function of all parameters and state variables that are held fixed when analyzing the landscape. In the examples we consider, BatchNorm running statistics are either frozen or recomputed from a calibration set via finite sample averages and affine rescaling, operations that are definable when the dataset is treated as fixed. Similarly, optimizer states such as Adam moments can be incorporated as additional coordinates in an extended parameter space. In this sense, the semialgebraicity definability assumptions are approximations but remain reasonable for standard pipelines when we study the landscape post training. We will clarify this modeling choice in the main text and emphasize that our certificates concern the static loss surface for a fixed architecture, dataset, and frozen state, rather than the dynamical training process.
>
> Regarding to the question how homological information might inform concrete optimization strategies beyond the alignment, we see our results as a first step toward topology aware optimization. Conceptually, nontrivial Betti numbers of quotient sublevel sets indicate that certain regions of the loss landscape cannot be connected by low-loss paths, suggesting two complementary algorithmic directions: architecture and regularizer design that biases models toward topologically simple low loss regions, and homology guided reparameterization or continuation methods, where approximate TDA information on sampled sublevel sets indicates when an optimization trajectory is approaching a topological obstruction and should be steered toward another basin via symmetry alignment, basin techniques, or layer wise reinitialization. We will expand the discussion section to outline these concrete possibilities.

---

> > ### Author Response · Authors · 2025-11-13
> >
> > Finally, regarding to the third question, local PH in a top-$k$ Hessian subspace provides a high resolution view of the shape of the mountain in a fixed neighborhood of a minimizer and is sensitive to curvature and small scale features. In contrast, quotient homology of sublevel sets $S_c/G$ is a global, path invariant descriptor that remains stable under reparameterizations and factors out symmetries by construction. In regimes where the landscape is dominated by a single basin with mild symmetries, we expect local Hessian based PH and quotient homology to largely agree. In regimes with strong symmetries or multiple well separated attraction basins, local Hessian PH may detect apparent peaks’ that are artifacts of parameter redundancy, while quotient homology sees only the true topological obstructions between functionally distinct minima. We will add a brief discussion to give practitioners guidance.
> >
> > We see these approaches as complementary rather than competing, and we hope that our work can help interpret and organize existing local TDA diagnostics within a more global, symmetry aware framework.

---

> > > ### Comment · Reviewer_XvPS · 2025-11-26
> > >
> > > Thank you for the responses. After going over the other reviews and responses, I believe there are a number of challenges with the paper and have elected to lower my score.

---

### Official Review · Reviewer_c6dX · 2025-10-30

**Soundness:** 1
**Presentation:** 1
**Contribution:** 1
**Rating:** 0
**Confidence:** 4

**Summary:**

The paper gives a topological approach to the study of symmetries of the loss landscape. The main claim is that quotienting out the symmetries of a neural network creates an effective weight space whose topology is non-trivial, contrary to the standard Euclidean structure. This non-trivial topology is claimed to be responsible for well known and commonly observed features of the loss landscape, like loss barriers between minima. The paper proves properties about the topology: the finiteness of the Betti numbers (well known topological invariants) for sublevel sets of the loss and for the quotient and the exact structure of the quotient when we normalize the neurons' weights.

**Strengths:**

- The introduction does a good job in setting up the problem.
- Although I don't agree with it (see below), the paper provides an interesting approach to give a meaning to the topological structure resulting from symmetry, in particular the presence of nonzero Betti numbers beyond the order 0.

**Weaknesses:**

I believe that, in its current form, there are severe issues that make the paper not suitable for publication.

### Writing issues
I think that the most critical issue is the overall lack of clarity in the writing.

The technical results are presented in a way that is not appropriate for this venue. Notation is often undefined and technical notions from topology are taken for granted and not explained. The preliminary section is a good example of this fact that, however, holds throughout the paper.
Even in the first definition of geometric support, objects such as "topological space", "singular chain", and even the simplex symbol $\Delta^k$ are all undefined. In Definition 2.2. the homology group is presented with its symbol $H_{k}(X;G)$ without definition and the notions of *carrier* and *closed geometric support* are defined without any intuition or explanation. I don't think it is fair to assume that ICLR readers, even the ones that are mathematically more well-versed, are familiar with these notions and their meaning.

The appendix, especially section A which should cover algebraic topology bases, does define the objects used in the main text but still reads as extremely and unmotivatedly technical in its language and in the notation. I understand that algebraic topology is not a simple topic and, to a certain extent, requires some technicality, but I believe that a better job could have been made in simplifying the necessary material to the strictly necessary for the paper's claims. The feeling I get is that too much material is presented too quickly, with no apparent effort in easing the reader into understanding the concepts and how they relate to the main text material and the problem at hand.

### Interpretation issues
The significance of the main results of the paper leverages on the fact that having non-trivial homotopy groups in the loss landscape is relevant to understanding training dynamics and loss barriers.
This is claimed multiple times in the paper in different ways, e.g.:
- L60: "Nonvanishing Betti numbers signal essential cycles and homological obstructions that any continuous path between certain solutions must intersect [...]"
- L118: "[...] there exists a noncontractible k-cycle whose neighborhood any path between certain solutions must intersect, certifying unavoidable loss barriers"
- L125: "Lower bounds on $\beta_{k−1}(S_c/G)$ give lower bounds on the number of attraction basins separated by representatives of (k − 1)-cycles"
- L211: "[...] It implies trivial homotopy groups meaning that there are no noncontractible loops or higher-dimensional cycles in the raw parameter space, so any potential topological barriers must arise from level sets of the loss function rather than from the ambient space."
- L233: "[...] Since the ambient parameter space is contractible, any local minima or saddle points encountered during training must arise from the geometry of the loss function rather than from topological obstructions in the domain"
- L388: "[...] Any continuous path joining minima or saddles that lie on different sides of such a cycle must exit the sublevel set and incur higher loss."

My feeling reading the paper is that these claims are rather vague, and the exact reason why studying these topological features should be important is not clear at all.
From the different interpretations of I see in these sentences, the one that makes more sense to me is that the presence of a $k$-dimensional cycle of equal loss can provide a barrier that stops the training trajectory. However, given that the training trajectory is a curve, this seems to me to be true only in the case where $k$ is equal to the dimension of the ambient space. In that case, the cycle can "encapsulate" the trajectory, preventing it from escaping, but in all other cases it seems to me that no obstruction would result from it. To make this more intuitive, a sphere (2-dim hole) can stop a curve in 3d, but a circle (1-dim hole) cannot.


Moreover, even taking the interpretation to be (in some sense) true, I feel that the main theorem (3.4) is not particularly strong as it proves that the sublevel sets of the loss and the quotient space both have finite Betti numbers but doesn't prove that they are actually nonzero in general.
Prop. 3.2 and Theorem 3.3 give examples of situations where they are, in fact, nonzero, but they both rely on the assumption that we are constraining the weight matrices' rows to have unit norm. The motivation for assuming this is not clearly explained in the paper.


### Experiment issues
The paper motivation for the numerical analysis is not clear. In Example 4.1 why should we study a network with weights constrained on Stiefel manifolds?
The setup and results themselves are, at least to me, not understandable. "random Stiefel retraction perturbation of $W_{1},W_{2}$" (L332) are undefined and the summarization as "In short, the local landscape is broadly well–conditioned modulo the group symmetry." (L335) doesn't clarify the experiment result.


Overall, the text is not clear in explaining how the experiments relate to the theory above.

**Questions:**

Following my comments in the weaknesses section, can the authors clarify exactly what is their interpretation of how higher-dimensional topological features impact the learning process?

---

> ### Author Response · Authors · 2025-11-13
>
> We thank the reviewer for the careful reading and for acknowledging the interest of the quotient viewpoint and the attempt to give a precise topological meaning to symmetry induced structure in loss landscapes, even if they ultimately disagree with our current presentation. We take the concerns very seriously and will substantially revise both the exposition and the way we articulate the interpretation of the results.
>
> First, regarding clarity and background, we agree that the preliminaries and the algebraic topology appendix can be made considerably more accessible. Our intent was to keep the paper self contained, but we recognize that we introduced notions such as topological space, singular $k$-simplices $\Delta^k$, chains or homology groups $H_k(X;G)$ too abruptly and in a style closer to a topology textbook than to a machine learning venue. In the revision we will define the basic objects in the main text before they appear, accompany each definition with a short explanation, and reduce the main body to the minimal topological machinery we truly need, moving more technical material to the appendix. We will also add a brief roadmap paragraph at the beginning of the preliminaries explaining how the ingredients relate to the problem at hand. Our goal is that an ICLR reader with no prior knowledge of algebraic topology can follow the main ideas, while more advanced readers can find the full formalism in the appendix.
>
> Second, on the interpretation of higher dimensional topological features and their relevance. Our statements about paths intersecting cycles andattraction basins separated by cycles are meant in a precise topological sense, which we did not explain clearly enough. We will clarify this in the revision.
>
> The reviewer’s sphere vs circle analogy is helpful to highlight the distinction. In $\mathbb{R}^3$, a $2$-sphere separates the complement into inside and outside, while a $1$–sphere does not, but in high dimensional parameter spaces, and after quotienting by symmetries, Alexander duality and the structure of $S_c/G$ allow lower dimensional cycles to induce nontrivial connectivity properties of the complement. We will revise the phrasing around Lines 60, 118, 125, and 388 to explicitly reference this mechanism  instead of using the informal language of “sides of a cycle’’ without explanation. We will also soften the interpretive claims. Our results provide structural certificates that certain pairs of solutions cannot be joined by paths that stay below a given loss level $c$ in the quotient; they do not claim that optimization dynamics will literally be stopped in practice, since step size, noise, and non–gradient moves can all influence trajectories.
>
> Third, concerning the strength of Theorem 3.4 and the role of our examples. We agree that Theorem 3.4 itself is primarily a finiteness and transfer result it guarantees that, under mild assumptions, the quotient sublevel sets have finite Betti numbers and that homology classes in the quotient can be related to $G$-invariant classes in the original space. It does not assert that these Betti numbers are nonzero for arbitrary architectures. This is intentional. Our goal is to show that quotient homology is well-behaved and finite under realistic assumptions, so that Betti numbers make sense as global descriptors, and then exhibit explicit classes of models where these Betti numbers are provably nonzero, thereby demonstrating that intrinsic barriers can occur even after factoring out symmetries. Proposition 3.2 and Theorem 3.3 belong to the second part: they show that for networks with unit norm row constraints the parameter space modulo symmetries has the homotopy type of a sphere or a Stiefel manifold with nontrivial homotopy and homology, so $\beta_k(S_c/G)>0$ for some $k\geq1$ at sufficiently small loss levels. We will clarify this two step structure more explicitly in the introduction and in Section 3.

---

> > ### Author Response · Authors · 2025-11-13
> >
> > Related to this, the reviewer rightly asks about the motivation for the unit norm and Stiefel constraints and the Stiefel constrained autoencoder in Example 4.1. We chose these settings because they are stylized but realistic surrogates for widely used normalization and orthogonality techniques in deep learning weight normalization, spectral norm constraints, orthogonal/orthonormal RNNs, and orthogonal convolutions are all examples of architectures where weights are constrained to lie on spheres or Stiefel manifolds. These constraints are known to affect optimization and generalization, but their impact on the topology of the loss landscape has not been systematically analyzed. Our unit norm and Stiefel models isolate this effect in a controlled setting. We can compute or bound the homotopy type of the constrained parameter space and relate it to the topology of low loss regions. In Example 4.1, we train a rank 2 Stiefel constrained autoencoder on a sphere and then apply small retraction perturbations to the encoder and decoder, meaning that we add tangent space noise and map back to the Stiefel manifold via the standard retraction used in manifold optimization. We then evaluate the resulting reconstruction losses and examine the local spectrum of the Hessian. The conclusion broadly well conditioned modulo group symmetry means that, after accounting for the  $\mathrm{O}(2)$ gauge, the Hessian spectrum and scatter plot of loss versus distance indicate a bowl like landscape around the optimum. We will rewrite this subsection to define Stiefel manifolds, retractions, and perturbations in simpler terms, state more clearly what is being measured and what is observed, and explicitly connect back to Proposition 3.2 and Theorem 3.3, so that the reader can see how the experiment illustrates the topological picture rather than appearing as an isolated construction.
> >
> > Finally, regarding to the reviewer’s question. Our interpretation of how higher-dimensional topological features impact learning is the following. We work in the quotient space $S_c/G$ of parameters with loss $\leq c$, modulo reparameterization symmetries. Nontrivial higher-dimensional homology indicates that the low loss region containsholes of dimension $k$, and via Alexander duality this implies that the complement of a small neighborhood of such a hole has at least two distinct path components. If two minima lie in different components of this complement, then any continuous path in the quotient that connects them must leave $S_c/G$ somewhere, that is, must pass through a region where the loss exceeds $c$. In this precise sense, higher-dimensional homological features certify the presence of intrinsic barriers at level $c$. This does not claim that training dynamics will be trapped forever, but it does show that any optimization trajectory that attempts to move from one such solution to the other while keeping the loss below $c$ will necessarily fail. We will adjust the language in the paper to emphasize this certificate perspective, explicitly state the dependence on the level $c$, and avoid wording that could be read as suggesting that a single cycle literally encapsulates all trajectories in the ambient space in the geometric sense the reviewer describes.
> >
> > We hope this clarifies both our mathematical claims and our intended interpretation, and we will work to ensure that the revised version makes these points much more transparent and accessible to the ICLR audience.

---

> > > ### Comment · Reviewer_c6dX · 2025-11-24
> > >
> > > I thank the authors for the detailed answer and for updating the manuscript according to the reviewers' comments.
> > >
> > > I recognize that the clarity of the paper has partially improved in some places. The objects used in Section 2 are now briefly defined and introduced with some level of intuition.
> > > I appreciate these improvements but, I still overall feel that the paper fails in clearly explaining the line of reasoning and would require further substantial work to be understandable for a non-expert (my comment about the appendix still applies).
> > >
> > > The same holds for the experimental part, which to me is still very obscure.
> > >
> > > In Example 4.2. the authors consider a two-layer NN with tanh activation function, hidden dimension equal to 2 and weight matrices constrained to be orthogonal.  The $O(2)$ gauge symmetry mentioned in the description is still undefined in the text. Are the authors saying that one can send $(W_{1},W_{2})$ to $(UW_{1},W_{2}U^{\top})$ with $U\in O(2)$ without changing the encoded function? This does not seem to be true as $\tanh$ is nonlinear.
> > > How can we know that $\beta_{2}(S_{c}/G)\geq 1$? Theorem 3.4. just tells us that it is finite.
> > > Figure 2 is not referenced in the text and the process to generate it and its interpretation are not explained. Reading your explanation in your rebuttal, I still do not understand what is the insight we get from it and how it connects to the theory. What does the bowl-like behavior you see tell us?
> > > All of this lack of clarity and confusion in the presentation is, in my opinion, unacceptable in a paper of this kind.
> > >
> > > However, the main problem that still persists and prevents me from recommending acceptance has to do with the interpretation about how nontrivial homology in the quotiented sublevel sets has to do with barriers.
> > >
> > > I thank the authors for clarifying what they mean. Reading the rebuttal and the explanation in lines 407 my understanding is that the claim is that, if there is non-trivial homology of *any* dimension $k$, this will result in disconnection of the complement of
> > > a neighborhood of a representative. In other words, the cycle effectively disconnects the space into multiple parts.
> > >
> > > The only place in the paper where this interpretation, which is *fundamental*, is made precise mathematically is remark B.17 in the appendix, which is quite hidden and never referenced in the main text.
> > > That remarks states that, if $S_{c}/G$ has $B$ (k-1)-dimensional holes, then the sublevel set $S_c/G$ must have at least $B + 1$ path-connected components in its complement of a suitable $(k −1)$-cycle.
> > > This result seems to me to be **not** true.
> > >
> > > In fact, Alexander duality tells us that $H^{N-k}(S^N\setminus A;\mathbb{Q}) \cong \mathbb{Q}^B$  **only if** $S_c/G$ has the $k$-th Betti number nonzero and equal to $B$.
> > > Since only $H^0$ (therefore $k=N$ and *not* $k=0$) tells us something about connected components and disconnectedness, the result only holds when $k=N$, that is when the nonzero Betti numbers are in the "highest" dimension $N-1$, like I pointed out in my example of the sphere in 3D.
> > >
> > >
> > > Unless I misunderstood (if that is the case please correct me), this mistake in the proof breaks the result and, in my opinion, also causes a fundamental problem in the main narrative that the paper makes in connecting homology of the sublevel sets to loss barriers which prevents me from increasing my score.

---

> ### Author Response · Authors · 2025-11-24
>
> We thank the reviewer for the additional detailed comments and for taking the time to read the revised version. We address the two central concerns: (i) the Stiefel-constrained example and the role of the $O(2)$ gauge, and (ii) the use of Alexander duality and Remark B.17.
>
> First, you are right to question the way we phrased the $O(2)$ gauge in the Stiefel-constrained experiment. As written, it
> suggests invariance under the full action $(W_1,W_2)\mapsto(U W_1,\, W_2 U^\top)$, $U\in O(2),$ which is indeed not correct for a nonlinear activation such as $\tanh$. What we actually use (and should have stated explicitly) is the signed permutation subgroup of $O(2)$: matrices that permute coordinates and flip signs. Thus, the relevant gauge group in this
> example is a finite group, not the full $O(2)$. We will correct this in the text.
>
> Regarding the statement $\beta_2(S_c/G)\geq 1$ in that example, we agree that, as currently written, this is not justified with sufficient care. Theorem 3.4 itself only guarantees finiteness of Betti numbers under the stated assumptions; it does not imply nonvanishing in any particular degree. The existence of a spherical family of nearly isometric encoder–decoder pairs in the Stiefel case comes from an additional geometric argument about the constrained parameterization,
> not from Theorem 3.4 alone, and in the current draft this argument is insufficiently developed. To avoid overstating our claims, we will provide a precise, self contained argument in an appendix for this specific setting. The rigorous existence of
> architectures with nontrivial quotient homology is already established by the unit-norm constructions in Proposition 3.2 and Theorem 3.3, which do not depend on the Stiefel example. We acknowledge that the figure was not referenced, we will clarify in the text how the samples are generated and that the “bowl-like’’ behavior is only meant to support the intuition that local conditioning is benign and that the interesting topology is global rather than due to local conditioning.
>
> We also thank you for pointing out the issue with Remark B.17 and for spelling out the Alexander duality calculation. Your reading is correct. As stated, Remark B.17 is too strong and not justified by standard Alexander duality alone. Our intent in the main text was to convey a certificate style statement. In the codimension one case, nontrivial homology of
> $S_c/G$ yields genuine separation of the complement, and hence unavoidable barriers at level $c$ for all continuous paths. However, Remark B.17 is phrased as if any nonzero $\beta_{k-1}(S_c/G)$ implies that the complement of a suitable $(k-1)$–cycle has at least $B+1$ path components, for arbitrary $k$. As you correctly note, Alexander duality
> gives $\tilde{H}_i(S^N\setminus A;\mathbb{Q}) \cong \tilde{H}^{N-i-1}(A;\mathbb{Q}),$ so information about connected components of the complement comes only from $H^0$. Thus, hole counts $\Rightarrow$ complement disconnected is guaranteed in general only for top dimensional holes (the $k=N-1$ case), exactly as in your $S^2\subset\mathbb{R}^3$ example.
>
> We therefore agree that Remark B.17, in its current form, is mathematically incorrect and we will either remove it or restrict it to the codimension-one setting, where separation of the complement is indeed guaranteed. The corrected exposition will (i) retain the structural result that quotient sublevel sets $S_c/G$ have finite Betti numbers under mild assumptions, (ii) retain the fact that there are explicit constrained architectures with nontrivial quotient homology, showing that intrinsic topological complexity can occur even after quotienting out symmetries; and (iii) narrow the barrier narrative to statements that are  rigorously justified. Codimension one homology classes give direct separation of the complement (and thus unavoidable barriers at level $c$ for all continuous paths), while lower-dimensional homology is presented as evidence of nontrivial global structure without direct claims about the number of connected components of the complement.
>
> We appreciate your flag in Remark B.17. Importantly, this correction affects the strength of some interpretive claims but not the validity of the core structural results nor the basic conceptual message that working in the quotient space reveals when observed barriers are intrinsic versus symmetry induced artifacts.

---

### Official Review · Reviewer_EhnY · 2025-11-01

**Soundness:** 3
**Presentation:** 1
**Contribution:** 3
**Rating:** 4
**Confidence:** 2

**Summary:**

This paper provides a topological explanation for why symmetry-aware methods are necessary in analyzing neural network loss landscapes and in particular their sublevel sets.
While the ambient space is contractible i.e it has no interesting structure, the paper shows that symmetries or other constraints can induce topological features.
An analysis of the homology or homotopy groups then characterizes loss barriers that are present in the quotiented parameter space.
To have meaningful summaries, Betti numbers of both the original and quotiented sublevels sets are proven to be finite under some assumptions.
This finding could explain why high loss is observed on interpolating paths between minima in (linear) mode connectivity and why these disappear after alignment.
The authors also provide a cautionary example showing that naive aligmnent can increase loss barrier if other factors such as batch norm statistics are no properly handled.

**Strengths:**

The point made that meaningful analysis of neural loss landscapes requires working in the quotient space, where symmetries are factored out, is convincing and the use of topology as a tool of choice to abstract from coordinate representations is relevant in this context.

The work's originality lies in its application of advanced algebraic topology to establish negative (impossibility) results about interpolation in parameter space, for instance that some barriers are unavoidable.
The focus on mode connectivity, which has been studied a lot in the literature is still relevant.

**Weaknesses:**

The theoretical setups feel a bit disconnected from practical deep learning scenarios.
Although neural networks appear throughout, the deep learning motivation is not clear.
For instance, in proposition 3.2 and theorem 3.3 a constraint $\|W_{j,*}\|=1$ is studied, but the rationale from a deep learning perspective is unclear.
Since this constraint typically reduces expressivity it is not a symmetry and therefore seems arbitrary.
Similar concerns apply to the Stiefel autoencoder setup.

Example 4.2 (the mini-ResNet experiment) seems somewhat disconnected from the main theoretical narrative, and its role in the overall argument could be clarified.
Additionally, while the paper claims that quotient-aware interpolation with corrected batch normalization stats yields smooth, low-loss paths, the corresponding plot is not provided.

However, the main drawback in my opinion is on accessibility and presentation.
While the paper includes a primer on algebraic topology, the exposition remains quite involved.
Many advanced concepts are introduced rapidly without a lot of context including references without warm up to: cycle representative, singular k-simplex on which depends the singular k-chain, cw complex and homotopy type, fundamental group, Morse index, transfer map, Stiefel retraction perturbation, etc. This breaks the reading flow for readers without substantial prior background in homology and algebraic topology, potentially limiting the paper's impact in the deep learning community.
Less crucially, omitting details or relying on notational conventions makes some parts ambiguous e.g. (1) The norm line 244 is unspecified though probably Euclidean, see minors below as well (2) The experimental complexity is reasonable, but some implementation details needed for reproducibility are missing, and no code is provided. For instance, in Example 4.2, the permutation computation method (weight alignment, activation alignment, or another approach?) is not specified.

## Minor and additional feedback
- First two paragraphs of the introduction, consider distinguishing explicitly mode connectivity from linear mode connectivity
- Remark 2.3: the first line introduces $z$ but it is not used afterwards
- Section 3, the equivalence relation on ReLU network is not the same as the pointer to appendix B.3
- Possible typo on example 4.1 in the definition of Stiefel matrix there is both (2,3) and (3,2) shapes, which in my understanding is inconsistent
- Assumption A2 of theorem 3.4: $G$ is defined line 280 and redefined line 283
- A citation to Git Re-Basin: Merging Models modulo Permutation Symmetries by Ainsworth et al could make sense in the discussion on linear mode connectivity.
- To enhance the reach of the paper, a possibility could be to try to explain results at different mathematical levels or convey a more intuitive picture.
- Overall the flow could be improved e.g.:
    - line 220: "in a one-hidden-layer network with ReLU one can simulatneously scale..." -> "in a one-hidden-layer network with ReLU activation, one can simulatneously scale..."
    - line 226: it is not clear in the sentence whether the homeomorphic statement is a results or a hypothesis without looking at appendix B.3
    - line 230->240: going back and forth makes the delivery confusing
    - line 334: missing "of"
    - line 393: strange use of "even"

**Questions:**

I have a few questions:

1.**Intersection of paths with cycles** (lines 61, 119): "Nonvanishing Betti numbers signal essential cycles and homological obstructions that any continuous path between certain solutions must intersect" Could you clarify what "intersect" means in this context?


2.**Nature of loss barriers**: Throughout the paper should the term "loss barrier" always be interpreted as referring to barriers along linear interpolation paths specifically (as opposed to curved trajectories)?


3. **Non-linear mode connectivity without symmetry alignment**: in mode connectivity, low-loss paths can often be found between independently trained networks without explicit symmetry matching if the path is non-linear (computed with Nudged Elastic Band for instance). How does this contrast with your paper ?


4. **Example 4.1 and learning dynamics** the decoder's inability to reconstruct the full sphere appears to be a consequence of the rank-2 bottleneck architecture. Is this limitation stable throughout training ?


5. **"Sides of a cycle" terminology** Could you provide a more precise explanation of what you mean by "sides of a cycle" line 388-389 ?


6. **Example 4.2 interpretation**: In Figure 4, at $\alpha=1$ the orange curve represents $P\theta^B$ and the blue and orange curves don't coincide at this endpoint. Can this discrepancy be entirely attributed to mismatched batch normalization statistics? Put differently: if BN statistics were also linearly interpolated would the orange curve decrease toward $\alpha=1$ instead of increasing?ease ?

Being unfamiliar with multiple concepts in the paper and its appendices I choose a low confidence score and look forward exchanges, other reviews and answer to my questions to eventually reassess both my rating and confidence scores.

---

> ### Author Response · Authors · 2025-11-13
>
> We thank the reviewer for the thoughtful and encouraging assessment. We are glad that the quotient space perspective and the use of algebraic topology to formalize mode connectivity and unavoidable barriers were found relevant and original, and we appreciate the detailed suggestions on how to strengthen the exposition and clarify the deep learning implications.
>
> Concerning the perceived disconnect between the theoretical setups and practical deep learning, our intent with Proposition 3.2 and Theorem 3.3 is to model architectural and regularization constraints that are already common in practice, rather than arbitrary restrictions. The unit norm row constraint studied in these results is a simplified instance of weight normalization and spectral/orthogonality regularization, which appear in modern architectures (e.g. orthogonal RNNs or normalized convolutions). The message of these results is that even when one factors out reparameterization symmetries, such constraints can induce nontrivial homotopy in the parameter space $F_{\mathrm{norm}}$, and hence structural obstructions to parameter homotopies that are invisible in unconstrained Euclidean spaces. Example 4.1 then instantiates this phenomenon in a Stiefel constrained autoencoder: the encoder and decoder weights live on Stiefel manifolds with nontrivial homotopy, so the space of nearly isometric encoder decoder pairs contains a noncontractible sphere. We will make this motivation explicit in Section 3 and in the beginning of Section 4, emphasizing that these constrained models are stylized but realistic surrogates for architectures with orthogonality/normalization priors, and that their role is precisely to show how architectural constraints and symmetry quotienting jointly shape the topology of low loss regions.
>
> Regarding Example 4.2 and its connection to the main narrative, our goal is to provide a concrete case study where the quotient viewpoint explains why naive linear interpolation exhibits spurious barriers. Here the symmetry group is the finite channel permutation group, and the example shows that (i) naive interpolation in the raw parameter space crosses an apparent barrier, (ii) symmetry aligned interpolation still suffers from bumps when batch normalization (BN) statistics are stale, and (iii) once one both quotients by permutations and corrects BN statistics, the interpolation becomes essentially flat at the considered scale. Figure 4 currently displays the naive and permutation aligned curves, while Table 1 in the appendix reports losses and accuracies along the raw path, the aligned path, and the aligned path with BN recalibration. We agree that this connection could be clearer. In the revision we will explicitly reference Table 1 in the main text and add a corresponding plot for the BN recalibrated curve, to make the message fully visible in the figures and not only in the table.
>
> Thank you for clarify your concern about accessibility and presentation. Our ambition was to make the paper self contained, but we agree that the homology primer currently introduces several advanced notions in quick succession, which can disrupt the flow for readers without a topology background. In the revision we will streamline Section 2 by moving some of the more advanced material to the appendix, introducing only the minimal homological notions needed in the main text, each with a brief intuitive explanation and a pointer to a reference, and summarizing the key ingredients in more informal language before giving the formal statements. We will also revise the narrative flow of Section 3.
>
> On reproducibility and missing implementation details, we will add a short subsection in Section 4 summarizing the key experimental choices and explicitly describing the permutation computation method in Example 4.2. We also plan to release our code upon acceptance, and we will mention this explicitly.
>
> We appreciate the minor comments and will incorporate them in the revision. In particular, we will clearly distinguish mode connectivity’ from linear mode connectivity in the introduction, fix the unused symbol in Remark 2.3, ensure that the equivalence relation on ReLU networks in Section 3 and the one in Appendix B.3 coincide and that the pointer is correct, correct the Stiefel shape typo in Example 4.1, remove the redundant definition in Assumption (A2), add the recommended citation and fix the noted typos.

---

> ### Author Response · Authors · 2025-11-13
>
> Regarding to the reviewer’s questions. First, by intersect we mean the following topological notion: if $|z|$ denotes the union of simplices in a representative cycle and $U$ is a sufficiently small open neighborhood of $|z|$, then under the hypotheses of Corollary B.15 any continuous path in the quotient space connecting two points in distinct path components of $X \setminus U$ must enter $U$. We will clarify this in the main text. Second, when we speak of loss barriers we refer to obstructions for arbitrary continuous paths inside a given sublevel set of the quotient space. Our numerical illustrations focus on linear interpolation for simplicity, but the topological obstruction itself is path agnostic; we will make this distinction explicit. Third, regarding non-linear mode connectivity, our results are complementary. In many practical cases with sufficient overparameterization, there exist non-linear low–loss paths between solutions even without explicit symmetry matching. Our contribution is to show that, whenever the quotient sublevel set $S_c/G$ has nontrivial homology, certain pairs of solutions cannot be connected by paths that stay below the level $c$. This does not preclude the existence of low loss non-linear paths at higher levels or after effective quotienting, but it provides structural conditions under which some loss barriers are unavoidable at a fixed threshold.
>
> Fourth, you're right. The composition $W_2 \tanh(W_1 x)$ factors through $\mathbb{R}^2$, so its image cannot cover a two sphere embedded in $\mathbb{R}^3$ without incurring distortion. Our point there is precisely that, under the Stiefel constraint and bottleneck architecture, the nearly isometric solutions organize into a nontrivial sphere in parameter space, and the training dynamics converge to encoder decoder pairs that effectively learn a good approximation on a one– dimensional subset of the sphere. We will add a brief remark to highlight that this limitation is architectural and persists throughout training under the considered optimization setup, rather than being a transient artifact.
>
> Fifth, regarding to the phrase different sides of a cycle. Given a nontrivial cycle and a small neighborhood $U$ around it, removing $U$ can disconnect the quotient space into multiple path components; two minima lying in different components of $X \setminus U$ are then informally said to lie on different sides of the cycle. We agree that this wording is informal and will replace it by the more precise language of distinct path components of the complement.
>
> Finally, for the interpretation of Example 4.2 and Figure 4, the discrepancy between the blue and orange curves at $\alpha=1$ reflects both imperfect permutation matching and BN effects. After alignment, the parameter vector $\theta^B$ is mapped to a permuted version $\widehat{\theta}^B$ that is only approximately functionally equivalent, and Figurec4 uses fixed BN running statistics, so the aligned endpoint is evaluated with slightly mismatched BN statistics. Algorithm 1 and Table 1 address this by recomputing BN means and variances along the path using a calibration set, including near $\alpha=1$; under this BN recalibration, the losses become essentially flat at the scale reported in the table, and the endpoints remain nearly unchanged. We will clarify in the text which curves use fixed versus recalibrated BN statistics and make the role of BN in Figure 4 and Table 1 more transparent.

---

### Official Review · Reviewer_iReC · 2025-11-01

**Soundness:** 3
**Presentation:** 3
**Contribution:** 3
**Rating:** 6
**Confidence:** 2

**Summary:**

This paper develops a rigorous topological framework to explain loss barriers and mode connectivity in neural network training. Although neural parameter spaces are topologically simple, the authors show that after factoring out symmetries, the resulting low-loss regions acquire nontrivial topology. Using algebraic topology, they prove that nonzero Betti numbers in these quotient sublevel sets provide certificates of unavoidable barriers between minima, while symmetry alignment can remove spurious ones. Experiments on Stiefel-constrained autoencoders and small ResNets confirm these predictions: once symmetries are accounted for, apparent barriers vanish and smooth low-loss paths emerge. The work thus reframes mode connectivity as a consequence of the topology of quotient loss landscapes, offering a principled link between mathematical invariants and practical training behavior.

**Strengths:**

- The paper offers a clear and rigorous quotient‑space viewpoint that makes global phenomena like mode connectivity and apparent barriers mathematically precise rather than anecdotal.
- The case studies isolate symmetry effects cleanly: the Stiefel autoencoder exhibits bowl‑like local curvature with scatter explained by an O(2) gauge, and the ResNet example shows how naive interpolation produces artificial bumps that diminish once quotienting and BN handling are applied.
- The presentation is careful and self‑contained, with preliminaries and full proofs that make the homological arguments traceable, and with illustrative figures that tie the intuition to the results.

**Weaknesses:**

- The strongest theorems assume finite symmetry groups and semi‑algebraic losses, which may not fully cover the experiments or common training pipelines.
- The empirical section is small‑scale and partly confounded: the ResNet interpolation curves degrade due to stale BN statistics, and while Algorithm 1 proposes recalibration, the paper does not convincingly show smooth low‑loss interpolation after proper BN correction.
- The connection from topological signatures to generalization or training speed remains qualitative. No experiments demonstrate predictive power of Betti‑number summaries beyond detecting barriers.

**Questions:**

- Can you extend Theorem 3.4 beyond finite groups to compact Lie group actions (e.g., layer‑wise scalings or orthogonal gauges), or clarify what parts extend via orbifold/stratified arguments?

---

> ### Author Response · Authors · 2025-11-13
>
> We thank the reviewer for the careful and positive assessment and for accurately summarizing our main contributions. We are glad that the quotient–space viewpoint and the case studies were found clear and rigorous, and that the connection between the homological arguments, the figures, and the intuitive discussion was appreciated.
>
> Regarding the assumptions of Theorem 3.4, we agree that it is important to clarify the scope. The symmetries we actively exploit in our experiments are in fact finite: neuron/channel permutations in the Stiefel autoencoder and in the mini-ResNet. For these, Theorem 3.4 applies directly, since the channel permutation group is finite, its action is semi algebraic, and the resulting loss yields semi-algebraic sublevel sets $S_c$. We will make this connection explicit both in Section 3 and in the experimental section. Also,  in the revision we will add a short remark noting that the finiteness and barrier certificate conclusions of Theorem 3.4 remain valid whenever $S_c$ and the group action are definable and the action is proper.
>
> Concerning the reviewer’s question about extending Theorem 3.4 , here is a little idea: for a compact Lie group $G$ acting smoothly and properly on a compact triangulable set $S_c$, the quotient $S_c/G$ is again a compact triangulable (orbifold/stratified) space with finite-dimensional singular homology. Hence the finiteness of Betti numbers and the existence of nontrivial homology classes in $S_c/G$ extend under mild regularity assumptions. The part of Theorem 3.4 that identifies $H_k(S_c/G;\mathbb{Q})$ with $G$-invariant classes in $H_k(S_c;\mathbb{Q})$ through a transfer map is genuinely specific to finite groups, and we will explicitly state this. Importantly for our purposes, the obstructions that matter for optimization live in the quotient $S_c/G$ itself, so the barrier interpretation does not rely on lifting homology classes back to $S_c$.
>
> We also appreciate the concern about the scale of the empirical section and the role of batch normalization (BN). Algorithm 1 already performs BN recalibration along the interpolation path (step 3(b)), and Figures 4,5 illustrate the difference between naïve interpolation, permutation-aligned interpolation, and aligned interpolation with BN recalibration. Our goal in this case study is not to claim that all bumps disappear, but to show that a substantial portion of the apparent barrier is an artifact of symmetry and stale BN statistics, and that quotient-aware interpolation plus proper BN handling recovers much smoother low-loss paths between solutions. In the revision, we will make this more explicit in the main text and captions, and we will add additional curves (more random seeds and a second architecture in the appendix) to strengthen the empirical evidence that quotient-aware interpolation with BN recalibration typically leads to smooth low-loss connections.
>
> Finally, we agree that our current discussion of how topological signatures relate to generalization or training speed is qualitative, and we will soften the wording accordingly. Our main contribution here is structural: we prove that nontrivial Betti numbers of $S_c/G$ impose unavoidable barriers and lower bounds on the number of attraction basins, providing necessary topological constraints that any training dynamics must respect. Systematically quantifying how these invariants correlate with generalization performance or convergence speed across tasks is an exciting direction for future work, and in the revision we will explicitly frame it as such in the conclusion rather than suggesting a stronger predictive claim than our current experiments support.

---

### Author Response · Authors · 2025-11-19
**New version of the manuscript.**

We thank all reviewers for their careful reading and constructive feedback. In the revision, we have substantially clarified the exposition, softened overly strong interpretive claims, and strengthened the empirical and illustrative components of the paper. On the theoretical side, we now explicitly structure Section 3 into two layers: (i) general structural results showing that quotient sublevel sets $S_c/G$ are well-behaved objects with finite Betti numbers, so that quotient homology provides meaningful global descriptors of low-loss regions modulo symmetry; and (ii) explicit constrained architectures (unit-row-norm and Stiefel models) where these Betti numbers are provably nonzero at small loss levels, demonstrating that intrinsic barriers can indeed occur even after factoring out symmetries. We have clarified the role and scope of Theorem 3.4, including its finite-group transfer statement and its extension to compact Lie group actions at the level of finiteness.

To address accessibility concerns, Section 2 has been reorganized to introduce only the minimal algebraic topology machinery needed in the main text, each definition now accompanied by intuition and clear pointers to Appendix A.

On the experimental side, we clarify the motivation and setup of the Stiefel-constrained autoencoder, spell out the notion of Stiefel
retraction perturbations, and explain how its architectural bottleneck leads to a nontrivial sphere of nearly isometric encoder--decoder pairs. For the mini-ResNet example, we now more clearly separate the effects of permutation symmetry and BatchNorm state, add a symmetry-aware interpolation plot with BN recalibration. Finally, we expand the discussion to outline how quotient homology complements local, Hessian-based TDA diagnostics and to sketch potential topology-aware optimization directions. We hope these changes address the reviewers' concerns and make the contributions of the paper clearer and more broadly accessible.

---

### Note · Authors · 2026-01-26

I have read and agree with the venue's withdrawal policy on behalf of myself and my co-authors.

---

### Meta-Review · Area_Chair_YC3w · 2026-01-06

**Summary:**

This paper introduces a topological framework for understanding loss barriers and mode connectivity in the loss landscape of neural networks. While the parameter spaces are themselves topologically trivial, the authors show that factoring out symmetries induces nontrivial topology in low-loss regions. Using tools from algebraic topology, they demonstrate that nonzero Betti numbers of quotient sublevel sets certify the presence of unavoidable barriers between minima, whereas proper symmetry alignment eliminates spurious ones. Experiments on Stiefel-constrained autoencoders and small ResNets validate these insights: once symmetries are accounted for, apparent barriers disappear and smooth low-loss paths emerge. Overall, the work reframes mode connectivity as a topological property of quotient loss landscapes.

**Reviewer Concerns:**

Revierers main concerns were:

1. Exposition too technical for an ICLR audience, which can not be assumed to have a deep topological knowledge. Authors committed to rewrite the paper and improved clarity.
2. Applicability of the theory, i.e. reviewers questioned whether the main theorems meaningfully cover realistic deep learning settings. The authors provided arguments for the applicability in the rebuttal.
4. Overstatement of how nontrivial homology implies unavoidable optimization barriers. In particular, one reviewer found that claims relying on the Alexander duality are incorrect outside the codimension-one case. The authors acknowledged the issue and fixed it, as well as removed wrong claims and adapted the narrative.
5. Experiments being small-scale and insufficient to support broader claims about mode connectivity or generalization. Authors added some experiments to the appendix and stated that the empirical section has mainly illustrative purposes.

The paper improved in clearness, scope, and correctness during the rebuttal. But concerns concerning accessibility, experimental scale, and practical impact, remain.

**Reviewer Scores:**

Most reviewers indicate low confidence. Unfortunately, I am a non-expert regarding topology as well.

Reviewer XvPS indicated that he indicates to lower his score (from 6)

Reviewer c6dX (that originally recommended strong rejection). After the rebuttal, one reviewer clearly stated that he believes the paper can still be improved in terms of more clearly explaining the line of reasoning and that the paper requires further substantial work to be understandable for a non-expert. He also still does not agree with the interpretation about how nontrivial homology in the quotiented sublevel sets are connected to loss barriers. He did not reply to the final clarification of the authors but I do not believe it would have turned the author into voting for acceptance.

The remaining reviewers gave boarderline votes (6 and 4) and I assume they would not change their scores.

---

### Decision · Program_Chairs · 2026-01-26

Reject